# Can phone surveys be representative in low- and middle-income countries? An application to Myanmar

Isabel Lambrecht[1], Joanna van Asselt [2]*, Derek Headey[3], Bart Minten[4], Patrick Meza[5], Moe Sabai[6], Thet Su Sun [5], Hnin Ei Win[6]

1 Development Strategy and Governance Division, International Food Policy Research Institute, Dushanbe, Tajikistan, 2 Development Strategy and Governance Division, International Food Policy Research Institute, Washington, DC, United States of America, 3 Development Strategy and Governance Division, International Food Policy Research Institute, Colombo, Sri Lanka, 4 Development Strategy and Governance Division, International Food Policy Research Institute, Vientiane, Laos, 5 Myanmar Survey Research, Yangon, Myanmar, 6 Development Strategy and Governance Division, International Food Policy Research Institute, Yangon, Myanmar

* j.vanasselt@cgiar.org

**Data Availability Statement:** All the rounds of the Myanmar Household Welfare data are available on our website https://myanmar.ifpri.info/2022/09/06/myanmar-household-welfare-survey-mhws-round-

## Abstract

For decades, in-person data collection has been the standard modality for nationally and sub-nationally representative socio-economic survey data in low- and middle-income countries. As the COVID-19 pandemic rendered in-person surveys impossible and unethical, the urgent need for rapid monitoring necessitated researchers and statistical agencies to turn to phone surveys. However, apart from pandemic-related factors, a variety of other reasons can render large segments of a population inaccessible for in-person surveys, including political instability, climatic shocks, and remoteness. Such circumstances currently prevail in Myanmar, a country facing civil conflict and political instability since the February 2021 military takeover. Moreover, Myanmar routinely experiences extreme weather events and is characterized by numerous inaccessible and remote regions due to its mountainous geography. We describe a novel approach to sample design and statistical weighting that has been successfully applied in Myanmar to obtain nationally and sub-nationally representative phone survey data. We use quota sampling and entropy weighting to obtain a better geographical distribution compared to recent in-person survey efforts, including reaching respondents in areas of active conflict. Moreover, we minimize biases towards certain household and respondent characteristics that are usually present in phone surveys, for example towards well-educated or wealthy households, or towards men or household heads as respondents. Finally, due to the rapidly changing political and economic situation in Myanmar in 2022, the need for frequent and swift monitoring was critical. We carried out our phone survey over four quarters in 2022, interviewing more than 12,000 respondents in less than three months each survey. A survey of this scale and pace, though generally of much shorter duration than in-person interviews, could only be possible on the phone. Our study proves the feasibility of collecting nationally and sub nationally representative phone survey data using a non-representative sample frame, which is critical for rapid monitoring in any volatile economy.

1/. They can also be found on Harvard Dataverse at
https://doi.org/10.7910/DVN/1R3F3U for round 1,
https://doi.org/10.7910/DVN/LPNMTK round 2,
https://doi.org/10.7910/DVN/GVJKAI round 3, and
https://doi.org/10.7910/DVN/IKGJWF, round 4.

**Funding:** This study was made possible by the
support of the American people through the United
States Agency of International Development
(USAID), under the terms of Agreement No.
72048221IO00002) and the Livelihoods and Food
Security Fund (LIFT) (Grant Number: R 1.4/029/
2014). The funders had no role in study design,
data collection and analysis, decision to publish, or
preparation of the manuscript.

**Competing interests:** The authors have declared
that no competing interests exist.

## Section 1: Introduction

Despite their ability to access respondents in hard-to-reach places, prior to COVID-19 the use of national phone surveys was limited in low- and middle-income countries (LMICs), though growing [1–5]. With the onset of COVID-19 the implementation of phone surveys grew tremendously, mostly through necessity rather than desire. Three years into the pandemic, phone surveys are no longer a rarity nor a novelty. Whereas sampling and weighting procedures for generating nationally and sub nationally representative datasets are well established for in-person survey data collection, there is no such gold–or even silver–standard (yet) for phone survey data collection and phone surveys often get scrutinized for representativeness.

One challenge when setting up a phone survey is getting access to a sampling frame, which for a phone survey is a comprehensive list of phone numbers. Researchers that did organize national-level phone surveys mainly relied on random-digit dialing (RDD) [6], accessed contact information from households who were part of pre-COVID in-person survey datasets [7], or accessed lists of phone numbers from telecom providers in the country [8].

There are genuine concerns that those answering phone surveys might not be representative of all households or individuals in the population. First, not everyone owns a phone. Phone ownership is strongly associated with a range of factors, including geographical as well as household- and individual-level characteristics. Urban and suburban residents, and those living in well-connected areas with working infrastructure, including cell towers and electricity, are more likely to own phones [6, 9]. Moreover, mobile phone owners are typically wealthier, more educated, younger, and more often male than non-owners [6, 9, 10]. Second, non-response and refusal rates are much higher in phone surveys compared to in-person surveys and may also be linked to certain characteristics. Poorer people, farmers, or women may be less likely to answer a phone and could therefore be under-represented when randomly calling phone numbers [10–12].

Random digit dialing or calling respondents from telecom provider lists likely necessitates large amounts of phone calls to find respondents from these specific subpopulations—especially if regions are diverse in population size, phone ownership or response rates—thereby raising the cost of the survey quite substantially [13]. Using contact information from in-person survey datasets may result in underrepresentation of certain geographies or segments of the population who were difficult to travel to in-person. Further, the use of sampling weights cannot always correct these biases to ensure representativeness–especially at the individual level [6, 7, 9, 12].

The objective of this paper is twofold. First, we highlight the potential of phone survey data collection as a modality to obtain nationally and sub nationally representative socio-economic data where face-to-face interviews are not feasible at a large scale. We demonstrate how phone surveys can be a powerful tool to collect socioeconomic data in fragile states. While there are obvious challenges with achieving representativeness of phone surveys, there are also underappreciated advantages of phone surveys in countries beset by remoteness, conflict, and pandemic conditions [5]. Second, the paper provides details on an innovative sample design and weighting strategy to minimize bias in characteristics typically associated with phone surveys, even when starting with a non-representative sample frame.

The example we bring is from a data collection effort in Myanmar. By early 2022, Myanmar was grappling not only with the economic and health consequences of two years of the COVID-19 pandemic, including its fourth wave (Omicron), but also facing severe political instability, armed conflict, and insecurity following the military takeover from the quasi-civilian government in February 2021. In such a difficult environment, accurate and frequent monitoring of the population's welfare, with sufficient granularity, is crucial for targeting scarce

resources for maximum impact and benefit to vulnerable populations. Yet, in-person nation-wide survey data collection in Myanmar was, and is, infeasible.

We therefore organized a large-scale quarterly phone survey, representative at the national level, the urban/rural level, and the state/region level, called the Myanmar Household Welfare Survey (MHWS). Whereas previous socioeconomic surveys in Myanmar failed to reach significant parts of the country due to conflict (and were therefore perhaps not truly representative), the phone-based MHWS managed to survey the majority of Myanmar's townships, many of which have not been surveyed in recent times (e.g. northern Rakhine), and many of which were experiencing acute conflict and significant COVID-19 restrictions. Our findings demonstrate that the pre- and post-survey methods we employed to address biases in phone surveys could serve as a valuable template for implementing sub-nationally representative surveys in other countries facing intricate governance and logistical challenges.

The paper is organized as follows. In section 2 we discuss the data and methods, including the sampling and weighting strategy for MHWS. Herein, we describe considerations made in the sample design, challenges in implementing data collection in relation to the sample, and calculations of the weights to reduce bias resulting from the composition of the final sample. In section 3 we report sample characteristics of the survey and compare MHWS sample characteristics to other recent nationally representative datasets. We also discuss attrition in our sample over the four rounds of data collection that took place in 2022. In section 4, we discuss our findings, the contributions of our study, as well as its strengths and limitations. We conclude in section 5.

## Section 2: Data and methods

The first round (round 1) of MHWS data was collected from 12,100 households between December 17th, 2021, and February 13th, 2022. This was followed by a second round (round 2) in April-June 2022, a third round (round 3) in July-August 2022 and a fourth round (round 4) in October-December 2022. The objective of the survey was to collect data on a wide range of household and individual welfare indicators–including wealth, unemployment, food security, diet quality, subjective wellbeing, and coping strategies (survey instruments of rounds 1–4 can be found in [14]). In each round, the same questions were asked to capture changes in the indicators over time. In the S1 Table in S1 File we compare the indicators we collected in MHWS to the indicators collected in the 2017 Myanmar Living Conditions Survey (MLCS) (S2 Table in S1 File), the most recent in-person nationally representative socio-economic dataset for Myanmar. Because phone surveys are required to be shorter in duration than in-person surveys, MHWS does not collect detailed household member information or detailed food and non-food expenditures.

The implementation of the MHWS occurred in collaboration with Myanmar Survey Research (MSR), a private survey research company based in Myanmar. Interviews were conducted by enumerators using the KoboCollect App, a software used for data collection that is compatible with Computer Assisted Telephone Interviews (CATI) software. Enumerators conducted the interviews at the MSR office. MSR initially provided respondents with the equivalent of $2.20 per round as an incentive (round 1 and round 2) but increased that to the equivalent of $3.50 per round for subsequent rounds. The incentive is provided as a phone credit to the phone number that is contacted, using online systems for mobile phone credit purchases.

To minimize measurement error a quality control team monitored the progress of data collection and was on stand-by to provide feedback to interviewers as needed. Moreover, interviews were recorded, and supervisors were assigned a random selection of one out of five

interviews on which to conduct live recording checks. In addition, the data team performed daily checks to screen for outliers and conflicting responses, which were rectified by listening to the audio recordings or by calling back the respondents for further clarifications.

The aim of the MHWS was to represent the population living in conventional households, similar to the usual target population of nationally representative datasets that collect data through face-to-face interviews. MHWS respondents could be any household member aged 18–74 years old. The lower limit of 18 years old was purposively chosen as childhood legally ends at 18 years old in Myanmar. The upper limit of 74 years old was chosen with the intention of minimizing communication and recall errors that we expected to be common among respondents of high age. Below we distinguish three parts of the sampling process: (i) the original pre-survey sample design, (ii) the MHWS sample, and (iii) the calculation of sampling weights.

## 2.1 The MHWS sample design

The MHWS intended to interview 12,790 respondents for the 15 States and Regions in Myanmar (S1 Fig in S1 File). The sample was distributed proportional to the population size in each State and Region based on the population data of the 2014 Census [15], but with a minimum of 240 respondents for the two states with the lowest population size (i.e., Kayah and Chin State) (S3 Table in S1 File). The MHWS sample design was similar to the 2015–2016 Myanmar Demographic and Health Survey (MDHS; 12,500 households) and the 2017 MLCS (13,730 households), both of which aimed to be representative of each state/region and of rural and urban areas of Myanmar as a whole. The 2019 Inter-Censal Survey (ICS) estimates that in 2019, 85.8 percent of households owned mobile phones, 82.2 percent of rural households, and 94.9 percent of urban households. At that time, Ayeyawaddy region had the lowest mobile phone ownership, 76.0 percent of households, while Yangon had the highest, 94.4 percent of households [16].

At the time we initiated the MHWS survey MSR owned a database of 280,274 phone numbers of adults who consented to be contacted for future participation in phone survey data collection, including geographical information of the township of residence of the respondent. The MSR database was established through a combination of random digit dialing and referrals. Furthermore, starting in 2015, MSR asked respondents who participated in their opinion polls and market research studies to provide consent for future survey work.

The first step in selecting phone numbers for interview was the development of a sampling frame or master phone number database. To create the master database all phone numbers were stratified at the township level, which is a smaller geographical unit within each state or region, numbering 330 in total. The final amount of phone numbers in the master database were proportional to the population size in each township, based on the information in the 2014 Census data, but contained four times the actual number of target interviews for each township (to account for non-response). The township proportionality structure of this master database was designed to minimize the risk of oversampling respondents who live predominantly in well-connected and wealthier townships. Without a deliberate attempt to achieve such a spatial spread, a random selection of phone numbers risks reaching respondents who are clustered in urban areas, in localities with better infrastructure, higher levels of asset ownership, and located in more connected geographical areas within a state. For Myanmar, not achieving a broad spatial spread could also mean not reaching townships that are under control of ethnic armed organizations (which are often either less connected or using phone numbers of neighboring countries). While we did not insist on having an exact proportional balance of interviews at the township level in the final sample of interviewed respondents, the survey company did strive to achieve such balance to the extent possible.

Another concern we tried to mitigate was the likely underrepresentation of women, rural residents, people with lower levels of education and farmers. Such underrepresentation is often reported in phone survey samples [7, 10]. We therefore set the following quotas at the State/Region level (S4 Table in S1 File):

1. Gender (female): Half of all respondents should be female.

2. Location (rural): Respondents with rural residences should be proportional to the population in conventional households based on the 2014 Myanmar Census Report.

3. Education (lower-educated): Respondents who completed at most primary school level. The quota was calculated based on the percentage of adults in conventional households aged 25 years and over by highest level of education completed based on the 2014 census data. This percentage was then adjusted downward first to correct for the age range of our respondents (18 to 74 years old) and thereafter to account for shifting age cohorts between the time of the census data collection (2014) and the start of our survey (2021).

4. Household livelihood (farming): Respondents living in a household where crops were harvested in the past 12 months. The share of farmer households was calculated based on the same question in the nationally representative 2017 MLCS, but an additional 5 percent buffer was added. This oversampling of households with farm livelihoods was primarily because they are a key group of interest, and we planned follow-up surveys specifically for farm households.

In practice, the approach adopted to achieve these quotas was as follows. After explaining the purpose of the study and obtaining informed consent, the respondent first answered survey screening questions related to the quotas (age, gender, location, education level and household livelihood). Based on this information, it was assessed whether the interview quotas for respondents with these characteristics were already met, and if so, the respondent was explained that s/he would not be interviewed at this time but may be contacted again in the future.

Enumerators were clearly instructed that any household member between 18 and 74 years old was eligible to be interviewed (i.e., they did not need to target the household head for the interview). If the initial respondent who picked up the phone was not eligible to be interviewed because they were either too young or too old, the interviewer asked to speak with another household member whose age was in the desired age range. At the same time, there was no instruction to the interviewers that the owner of the phone number him- or herself should respond to the interview questions. In some cases, another person answered the call and agreed to be interviewed, while in other cases, the person who answered the call handed over the phone to another household member to be interviewed. This approach of respondent selection purposively deviated from the deliberate targeting of a knowledgeable adult, which is commonly done in household surveys. The intention was to ensure a better representation of the entire adult population in our sample, regardless of the persons' status in the household.

Sample deviations were expected given that a large share of the population was directly or indirectly affected by conflict, disruptions to telecommunication services, frequent power outages, economic distress, and displacement during the periods of data collection. While the final sample in round 1 did not fully achieve the attempted sample quotas and sizes, in round 4, the sample sizes and quotas were met (S5 Table in S1 File). In states and regions where MSR could not reach the targeted sample size adhering to the quotas after reaching out to all phone numbers in the master dataset, the survey company attempted to reach respondents from the respective townships in their database who were not selected in the master dataset. Even so,

attempted sample size targets and quotas could not always be met. The most severe problems of falling short of pre-determined sample size targets in round 1 were related to two issues. First, sample size gaps occurred in the two smallest states (Chin and Kayah). Reaching the targeted sample sizes there was more challenging than in other places due to a combination of higher target sample sizes relative to the population size (i.e., the intended oversampling), the remoteness of these areas (particularly in Chin State), and the fact that these areas were highly affected by active conflict. Second, it proved difficult to reach the quota of respondents with low levels of education in several States and Regions. Despite these shortcomings in round 1, in round 4 we slightly altered our data collection method, and as a result we better met our quotas and desired sample sizes.

From the beginning, enumerators were required to call a respondent five times at different times of day before dropping the household from the sample. In rounds 1, 2, and 3, we did not offer explicit descriptions of the amount of time to wait between 1) phone calls, and 2) the number of respondents called in each state/region. In round 4, we required enumerators to 1) wait at least 3 days between phone calls to the same respondent, and 2) stratify calls to each state/region over the three-months of data collection, with 25 percent of phone calls per state/region per month. This consistency allowed us to exceed our sample targets for Kayah and Chin, as well as meet our low-education target.

## 2.2 The construction of MHWS sampling weights

Two main steps were used to construct sampling weights: (a) we calculated a set of base weights that makes the required adjustments related to the sample target characteristics; (b) we applied a maximum entropy approach to further minimize residual bias in observed characteristics in terms of wealth and household composition. We re-calculate these weights in each round, following the strategy laid out below, to ensure representativeness of the data in each round.

**2.2.1 Base household weights.** For sample estimates to be representative of the population we developed the base household-level weights using three main steps:

1. Apply an expansion factor: We weight households for their probability of occurring in the sample. This step ensures representativeness at the State/Region level and the share of households in rural (urban) locations in each of these States and Regions. This can be written as follows:

$$hh\ weight\ v0_{is} = \frac{n_s}{N_s} \tag{1}$$

In which $n_s$ is the number of households in each strata s (i.e., urban or rural location of each State/Region as in the 2019 ICS) and $N_s$ is the number of observations in each strata.

2. Adjust for oversampling of farm households: In rural areas of each state and region we proportionally adjust the household weight according to the farm or non-farm status $f$ of households to have the same percentage $perc$ of farm households as found based on MLCS estimates in each rural strata $s$ as follows:

$$hh\ weight\ v1_{is} = hh\ weight\ v0_{is} * \frac{perc\ MLCS_{fs}}{perc\ MHWS_{fs}} \quad if\ rural = 1 \tag{2}$$

In (2), *perc MHWS$_{fs}$* was calculated using *hh weight v0$_{is}$*. No further correction for livelihoods was made at the urban level given the low number of farmers in that category.

3. Weight for education level of the respondent: We proportionally re-weight households to ensure that we achieve a similar percentage *perc* in each strata *s* of respondents with education level *e* (i.e., to adjust for oversampling of more educated respondents), as follows:

$$\text{base hh weight}_{is} = \text{hh weight } v1_{is} * \frac{\text{perc MLCS}_{ehfs}}{\text{perc MHWS}_{ehfs}} \tag{3}$$

Because there were significant differences between educational attainment by relation to the household head, location, and livelihood in the MLCS, weighting factors for step (3) were calculated based on the share of adults with education level *e* (low education or high education level) aged 13–69 years old in 2017 (i.e., who would be 18–74 years old in 2022), by relation to the household head *h* (head and spouse, versus other household members), by household farm or non-farm livelihood *f*, and by strata *s* (i.e. urban/rural location in each State or Region). Analyses of MLCS show no significant difference between the share of men and women who have low educational attainment, so weighting based on gender of the respondent did not seem warranted. In (3), *perc MHWS*$_{ehfs}$ was calculated using *hh weight v1*$_{is}$.

**2.2.2 Entropy-adjusted household weights.** Further bias in our sample may appear in terms of characteristics that are independent of the characteristics already stratified and weighted for. To correct for this kind of residual bias, we rely on the maximum entropy approach [17, 18]. This approach can be used to generate or adjust weights to match averages and totals of pre-selected indicators. It is increasingly being used to calibrate survey data to various population totals [17]. Land was also used to calibrate survey weights of several high frequency phone surveys initiated during the COVID-19 pandemic [19].

Calibration of the survey weights should only be based on characteristics that are time-invariant or slowly changing over time. The most recent nationally representative dataset (partially) available to these authors was the 2017 MLCS data, collected five years prior to the current MHWS. In a country that was transforming extremely rapidly prior to 2020 (including for example a massive surge in mobile phone ownership), and then was set back by a series of extreme social and economic shocks (i.e., a pandemic and conflict), many characteristics have likely changed. Moreover, a phone survey is typically short in nature and thus also limited in terms of available indicators to match between datasets. Moreover, while the 2019 Intercensal Survey (ICS) data was collected right before the onset of the pandemic, we can only access its reports but not the raw data itself.

The maximum entropy procedure is applied using the base weights calculated in step (a) and included constraints to maintain the total number of households in each State or Region and by urban and rural location (based on the 2020 estimate). Two additional sets of constraints were added related to wealth: (i) agricultural land owned (in five categories) among rural households, based on the distribution of the 2017 MLCS data; and (ii) housing type among urban households (apartment, bungalow/house, semi-pucca house, or other), based on the reported 2019 ICS information. Finally, we set constraints for household composition. More specifically, this approach adjusts for households where all adults are women (women-adult-only households, WAH) in rural and urban areas separately, based on the 2017 MLCS data.

Finally, we also develop population weights and individual weights. The population weights are calculated as the household weights multiplied by the number of household members reported by each respondent. Individual weights are developed for representation of individual-level data among the adult population (aged 18–74 years old). Individual weights are therefore also straightforwardly calculated as household weights multiplied by the number of adults in the household.

## 2.3 Ethics

All interviews were conducted by phone with respondents who consented to be interviewed during earlier recruitment by the survey company. Each phone call started with a short introduction of the study and informed consent. In addition, a screening question inquiring about the age of the respondent was asked to ensure no minors were interviewed. A written institutional ethics approval, IRB #00007490, was obtained from the International Food Policy Research Institute's institutional review committee prior to piloting and implementing the survey. The study was found to present no more than minimal risks to human subjects.

## 2.4 Demographic and socioeconomic comparability to previous surveys

We assess the performance of the dataset and sample weights in reflecting the spatial and socio-economic diversity of the country through comparisons with findings from three sources of survey data; (1) the 2015–2016 MDHS; (2) the 2017 MLCS; and (3) the 2019 ICS. Even though MDHS data are less recent than the other datasets, the MDHS is known as the national socioeconomic survey most successfully reaching parts of the country that are difficult to access. Hence, it is interesting to compare MHWS with MDHS in terms of spatial achievement. The publicly available MDHS dataset does not include variables allowing us to identify the township of the respondent and we therefore rely solely on the information available in their reports.

The MLCS data was the last nationally and sub nationally representative socioeconomic survey conducted in-person in Myanmar [20, 21]. This is the only dataset that allows us to identify the township where each household resides. The 2019 Inter-Censal Survey (ICS) is the most recent and relevant representative nationwide survey effort (December 2019 –February 2020), yet we do not have access to the dataset itself and thus are restricted to the information available in the ICS report [16]. Information on conflict is taken from ACLED (https://acleddata.com) and includes the number of battles, explosions, and violent events.

We compare unweighted MHWS sample means, weighted MHWS estimates and other nationally-representative (ICS or MLCS) household-level and individual-level estimates to explore biases in wealth and socio-economic status, and residual differences with other nationwide estimates after weighting. ICS data was collected during the same time of year as the first round of MHWS, which is relevant for comparing indicators affected by seasonality, such as the sources of drinking water. We similarly compare individual-level estimates of adults' ages, roles in the household and education with the MLCS dataset. Note that options to explore concerns related to biases in the MHWS is somewhat limited given the paucity of indicators that are expected to be similar and that are available in both datasets.

## Section 3: Results

### 3.1 Data collection in conflict-affected areas

The geographical coverage of the MHWS is more comprehensive in comparison to former national surveys. We use townships as an indicator of the geographical spread of the data. In round 1 respondents were reached in 310 of the 330 townships in Myanmar *(de facto* only 324, as we were legally restricted to excluding six townships in Wa SAZs). Fig 1 shows a map of respondents interviewed in round 1, whereas appendix S2 Fig in S1 File contains a map of enumeration for round 4. By round 4, the number of townships represented declined to 303. Between R1 and R4 conflict intensified across the country. At the same time, blackouts increased in frequency and duration. The list of the 20 non-surveyed townships in round 1 consists of excluded townships (six townships in Wa SAZs), townships with very low

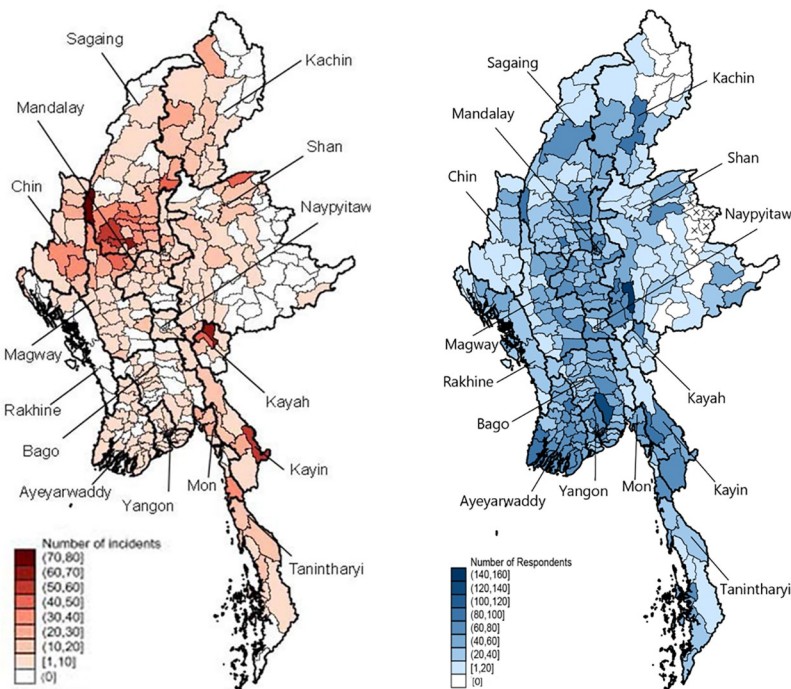

**Fig 1. Interviews conducted in the first round of MHWS (right) and conflict events taking place during the months of data collection (left), by township.** Note: stars indicate townships in Wa SAZ which were avoided for interviewing. Source: Reprinted from [MAPSA 2022] under a CC BY license, with permission from [Moe Sabai], original copyright [2022].

population sizes, or townships that are highly inaccessible–even by phone (Fig 1 and S6 Table in S1 File). In total the population of these townships consists of about 1.6 percent of the total 2019 population in conventional households of Myanmar, but about half of this non-enumerated population is from the excluded townships in Wa SAZ (Shan State). Hence from the perspective of the target survey population (i.e., excluding Wa SAZ), in R1 we only fail to reach townships that contain as little as 0.76 percent of the target survey population (S6 Table in S1 File).

While in both round 1 and round 2 we surveyed 310 townships, in round 3, three additional townships were not interviewed, all of which have very low population sizes. In round 4, three additional townships were not enumerated, this time due to intense fighting in Sagaing Region and Kayah State. Therefore, despite ongoing intense conflict, after four rounds, we only failed to reach townships that contain 1.46 percent of the target survey population (S6 Table in S1 File).

The MHWS geographical spread of 310 townships is better than face-to-face national-level survey efforts of similar sample sizes, such as the 2015–16 MDHS that reached 250 out of the 413 townships classified at that time [22] and the 2017 MLCS that reached 296 townships of 330 townships. See S3, S4 Figs in S1 File and S6 Table in S1 File for details.

Fig 1 compares a map indicating violent events taking place during the period of MHWS R1 data collection (based on ACLED data) with MHWS respondents reached in each township. S2 Fig in S1 File in the supporting information contains the maps for round 4 of MHWS. This comparison shows how we still managed to reach a reasonable number of respondents even in most of the conflict-affected townships. MHWS successfully reached

respondents in all 17 townships of Rakhine State. Fewer conflict events were experienced in Rakhine State in the recent year compared to the years prior, but nevertheless insecurity did not cease entirely.

We further compare the relationship between conflict incidence and conflict intensity with enumeration intensity for MHWS and MLCS in Table 1 and S7 Table in S1 File. Conflict intensity was measured through the monthly average of conflict events in non-enumerated townships during each survey's respective enumeration period and expressed relative to the population size (per 10,000 households). For each survey, we distinguish between underrepresented townships (≤ 7 respondents / 10,000 households), normally represented townships (7–14 respondents / 10,000 households), overrepresented townships (>14–22 respondents / 10,000 households) and largely overrepresented townships (>22 respondents/ 10,000 households). Note that a perfectly proportional distribution (obtained by dividing the total sample size by the number of households in the country) suggests 12 enumerated households per 10,000 households in MLCS and 11 enumerated households per 10,000 households in MHWS.

We find no patterns of exclusion or under-representation of townships who suffered from conflict in MHWS, but we do find this in the MLCS. Non-enumerated MHWS townships experienced similar or lower conflict incidence and intensity than enumerated townships (Table 1). During R1 of MHWS only one out of twenty non-enumerated townships experienced conflict. This is much lower than the national average, given that nationwide 239 of the 330 townships suffered from conflict in that period. In contrast, conflict incidence and intensity are much higher in MLCS non-enumerated townships compared to MLCS enumerated townships. Eleven out of 35 non-enumerated townships experienced conflict, whereas only 27 of the remaining 297 enumerated townships did so. In MLCS, because of security and access problems, 33 enumeration areas had to be replaced and seven were not visited at all [21]. This includes three townships in Rakhine state that could not be interviewed due to security concerns over the 12-month enumeration period.

Table 1 and S4 Fig in S1 File also demonstrate that MHWS reaches more townships and achieves a more even spread (i.e., less townships with no respondents and less townships with a disproportionally high amount of respondents). The twelve townships with higher-than-average representation in MHWS include four townships in Chin State and four townships in Kayah State. These States were intentionally oversampled given the small population size in these States, yet they are also two States highly affected by conflict. The MLCS similarly disproportionally interviews households from all townships in Chin and Kayah State, likely due to a combination of intentionally oversampling the smallest states, potentially aggravated by the clustered sample design.

**Table 1. Monthly average of violent events and experience of any violence during survey period, at township level and relative to population size.**

| Township groups based on number of households interviewed per 10,000 households | MHWS | | | MLCS | | |
|---|---|---|---|---|---|---|
| | # conflicts per 10,000 households | % townships with conflict | # townships | # conflicts per 10,000 households | % townships with conflict | # townships |
| No respondents (0) | 1.40 | 18% | 23 | 2.38 | 31% | 35 |
| Underrepresented (≤7) | 1.20 | 62% | 39 | 0.00 | 1% | 69 |
| Normal representation (7–14) | 0.79 | 72% | 212 | 0.07 | 7% | 134 |
| Overrepresented (14–22) | 1.42 | 79% | 43 | 0.28 | 19% | 31 |
| Largely overrepresented (>22) | 3.14 | 81% | 12 | 0.25 | 18% | 60 |

Note: One area was reclassified from one to two townships recently and we keep them together for this comparison table.

Source: the authors' estimates from MHWS, MLCS and ACLED (number of battles, explosions, and violent events)

## 3.2 Household composition and wealth: Comparisons with other nationally representative data

The household composition in MHWS is similar to the composition observed in MLCS. Table 2 shows the comparison of these characteristics from the pooled sample, and the accompanying appendix S8 Table in S1 File demonstrates the differences between unweighted and weighted MHWS estimates. We find a modest difference with MHWS households having more adults aged 15–64 years old, and fewer children and senior adults. Even though half of the phone survey respondents were female, the phone survey sample consisted of only 6 percent women-adult-only households (WAH) instead of 9 percent as in the MLCS dataset. However, after weighting, the share of WAH households increased to 9 percent, the same as in the MLCS.

Furthermore, the wealth indicators available from MHWS closely match those of MLCS and ICS, as shown in Table 3. It is possible that some differences could be explained by subtle differences in phrasing and administration of survey questions, but this caveat aside, the set of indicators we present here are broadly comparable. The MHWS estimates approximate those for ICS in most housing characteristics–including those that we did not control for in the weighting strategy (i.e., tenure of dwelling, type of floor and improved source of drinking water). Compared to the ICS, estimates from the MHWS show a slightly lower prevalence of households with an owned or freely provided house, fewer households with an improved floor, and fewer households with improved sources of drinking water. This suggests that the MHWS survey, if anything, would be biased towards less rather than more wealthy households, although we could be picking up some of the genuine welfare losses incurred by the severe economic shocks of 2020, 2021, and 2022.

A few points are also noteworthy from the weighted and unweighted comparisons reported in S8 Table in S1 File. First, comparisons of unweighted and weighted MHWS indicators shows that MHWS likely interviewed too many households who own larger areas of agricultural land, potentially indicating a bias in our sample towards more wealthy farm households that was not corrected for by our sampling targets (e.g., education). Second, urban sample respondents less often reside in apartments (11 percent versus 17 percent in ICS) and more often reside in wood/bamboo houses (55 percent versus 47 percent in ICS). This indicates a

**Table 2. Household composition comparisons between the MHWS and the MLCS.**

|  | National | | Rural | | Urban | |
|---|---|---|---|---|---|---|
|  | **MHWS** | **MLCS** | **MHWS** | **MLCS** | **MHWS** | **MLCS** |
| child <5y | 24% | 29% | 22% | 25% | 24% | 30% |
| child 5-14y | 48% | 51% | 44% | 44% | 50% | 53% |
| adult 15-64y | 100% | 97% | 100% | 97% | 99% | 97% |
| senior > = 65y | 22% | 25% | 21% | 27% | 23% | 24% |
| # children <5y | 0.27 | 0.34 | 0.25 | 0.29 | 0.28 | 0.35 |
| # children 5-14y | 0.70 | 0.80 | 0.62 | 0.67 | 0.72 | 0.85 |
| # adults 15-64y | 3.06 | 2.82 | 3.12 | 2.93 | 3.04 | 2.78 |
| # seniors > = 65y | 0.28 | 0.32 | 0.25 | 0.35 | 0.29 | 0.30 |
| Women adults only [a] | 9% | 9% | 10% | 10% | 9% | 9% |

[a] These are households with women adults (aged >14 years old) but without male adults (aged >14 years) in the household.

Notes: the estimates used the pooled sample from R1-R4, which has 49,294 observations.

Source: the authors' estimates from MLCS and MHWS.

**Table 3. Wealth indicators comparing MHWS (pooled sample and weighted) and ICS and MLCS survey findings, in percentage of households.**

| | National | | Urban | | Rural | |
|---|---|---|---|---|---|---|
| | MHWS | MLCS/ ICS | MHWS | MLCS/ ICS | MHWS | MLCS/ ICS |
| Agricultural land owned[a] | | | | | | |
| 0 acre | 63 | 63 | 90 | 92 | 53 | 51 |
| ]0–2] acre | 10 | 10 | 4 | 3 | 12 | 13 |
| ]2–4] acre | 8 | 9 | 2 | 2 | 10 | 12 |
| ]4–7.5] acre | 9 | 8 | 2 | 2 | 11 | 11 |
| >7.5 acre | 10 | 10 | 2 | 2 | 13 | 13 |
| Type of dwelling[b] | | | | | | |
| Wood/bamboo | 67 | 67 | 48 | 47 | 74 | 75 |
| Semi-pucca house | 14 | 13 | 18 | 18 | 12 | 11 |
| Bungalow | 12 | 11 | 16 | 16 | 10 | 8 |
| Apartment | 5 | 6 | 17 | 17 | 1 | 1 |
| Hut (2-3y) | 2 | 3 | 1 | 1 | 2 | 4 |
| Hut (1y) | 1 | 1 | 0 | 0 | 1 | 1 |
| House ownership | | | | | | |
| Owned/free | 90 | 93 | 74 | 81 | 96 | 98 |
| Rented | 9 | 7 | 25 | 19 | 3 | 2 |
| Squatter | 0 | n/a | 1 | n/a | 0 | n/a |
| Camp, shelter | 0 | n/a | 0 | n/a | 0 | n/a |
| Improved floor | 71 | 78 | 79 | 90 | 68 | 74 |
| Improved drinking water source | 78 | 82 | 93 | 92 | 72 | 78 |

[a] The comparison dataset for farm size is MLCS

[b] The comparison for housing indicators is the ICS.

Note: the estimates used the pooled sample from R1-R4, which has 49,294 observations.

Source: DoP and UNFPA (2020), and the authors' estimates from MLCS and MHWS.

potential bias in our urban sample towards fewer wealthy households. We corrected for these indicators in our weighting strategy, so this bias no longer appears in the weighted estimates.

### 3.3 Are respondents representative of the adult population?

Given that several indicators include individual-level—rather than household-level—characteristics, it is relevant to question the representativeness of the respondents for Myanmar's adult population. Table 4 shows comparisons of education level, age, relation to household head and gender between the adult-level weighted MHWS estimates, and the weighted 2017 MLCS adult population. Additionally, the comparison of unweighted and weighted MHWS estimates is shown in appendix S9 Table in S1 File. We assume that using the information captured in the household roster in a well-conducted national phone survey, such as the MLCS, allows one to confidently estimate key characteristics of all individuals living in conventional households (provided also that the correct weights are applied and ignoring for the moment the potential issues of non-representativeness of the MLCS).

We find a reasonable approximation of basic respondent characteristics of MHWS compared to MLCS in terms of education level, age, the relation to the household head and gender (Table 4). Our purposive strategy of interviewing adult household members that were not necessarily the most knowledgeable respondent or household heads, ensured to a large extent that our dataset does not suffer the same shortcomings as in other phone survey samples [7, 10],

**Table 4. Adult population (aged 18–74 years) characteristics in the MHWS (weighted) and MLCS (weighted), in percentage.**

| | National | | Urban | | Rural | |
|---|---|---|---|---|---|---|
| | MHWS | MLCS | MHWS | MLCS | MHWS | MLCS |
| low education level | 58 | 58 | 36 | 38 | 67 | 67 |
| junior: 18–24 | 18 | 17 | 22 | 17 | 17 | 17 |
| middle: 25-49y | 60 | 53 | 62 | 53 | 59 | 54 |
| senior: 50-74y | 22 | 30 | 16 | 30 | 24 | 29 |
| head | 34 | 34 | 30 | 32 | 35 | 34 |
| Spouse of head | 23 | 25 | 21 | 23 | 24 | 26 |
| Child of head | 33 | 34 | 37 | 36 | 31 | 34 |
| Other relation | 5 | 7 | 8 | 9 | 4 | 5 |
| Female | 53 | 54 | 53 | 55 | 53 | 54 |

Notes: The MHWS estimates used the pooled sample from R1-R4, which has 49,294 observations. Demographic variables from the MLCS dataset are based on the information from all household members aged 18–74 years included in the household roster.

Source: Authors' estimates from 2022 MHWS and 2017 MLCS

where respondents are disproportionally male and household heads. The share of youth in our sample is a good approximation of the share of youth in the general population, contrary to findings from other phone survey studies who find either an overrepresentation [23] or underrepresentation of youth [7]. However, we find a higher share of middle-aged people (25–49 years old) and a lower share of older people (age category 50–74 years old). This bias is particularly present among urban respondents and to a lesser extent among rural respondents.

S9 Table in S1 File shows that differences between adult respondents in the MHWS and the adult population in the MLCS further reduce after weighting the MHWS data. Despite setting (but not fully achieving) sample targets for respondents with lower levels of education, we observe an overrepresentation of respondents with higher education levels in the sample. However, this issue is resolved after weighting given that our weights are calibrated for education. The individual-level weights for the respondents are constructed by multiplying the household weights with the number of individuals (in our case, eligible adults) in the household, similar to how they are commonly derived for in-person household surveys. In contrast, such approach was not satisfactory for individual representativeness in several COVID-19 phone surveys in Sub-Saharan Africa [7], and further recalibration of phone survey sampling weights with propensity score adjustments (based on the individual's likelihood to be interviewed) was necessary to ensure that their individual-level characteristics would be closer to those of the benchmark population.

## 3.4 Attrition

The MHWS survey was initially designed to be a panel survey, but inevitably suffered from attrition. Between round 1 (December 2021- February 2022) and round 4 (October-December 2022) of the data collection, the spread and intensity of conflict, electricity blackouts, damage to telecom towers and purposive interruptions to mobile phone connectivity increased, and as a result, non-response was high among our previously interviewed respondents. Further, during the same period many households were displaced or fled the country. UNHCR (2023) estimates that between February 2021 and December 2022 1.14 million people in Myanmar were displaced internally because of conflict and 49,800 million individuals became refugees or asylum seekers in neighboring countries [24].

To replace the respondents that dropped out of the survey, the survey team called new respondents. The respondents were selected randomly from the phone database, in the same townships as the attrition respondents, and retained if they had similar characteristics to the attrition respondents in terms of urban/rural, gender, farm, and low education. If the survey team could not meet the quotas of certain target characteristics within the township, they called respondents that met these characteristics from the same state/region.

Thirty-six percent of respondents in round 1 could not be interviewed again in round 2. Yet, some of these were interviewed again in later rounds, thus only 31 percent of round 1 respondents were interviewed only once (Table 5 and S10 Table in S1 File). Moreover, the attrition rate decreased in the following rounds. S11 Table in S1 File shows in detail when respondents dropped out and re-entered the panel. Twenty-five percent of round 2 respondents could not be interviewed in round 3, and only 11 percent of them were interviewed only once across the four survey rounds. Similarly, 24 percent of round 3 respondents could not be interviewed in round 4, and only seven percent of round 3 households were interviewed only once (i.e., many of them were interviewed in the preceding rounds 1 or 2).

In Table 6, we explore differences between households that dropped out of the sample for each round (attrition households) and households that remained in the sample (panel households). We also investigate the differences between attrition households and replacement households. We present the marginal effects from probit regressions that explore the association of individual, household, and geographic characteristics on whether a household left the sample. In column (1) we analyze attrition from round 1 to round 2 by using round 1 characteristics and a dummy outcome variable for whether the household is still in the sample in round 2 (1 if attrition, 0 if still in the sample). In column (2) we analyze attrition from round 2 to round 3 by using round 2 characteristics and a dummy for if the household is still in the sample in round 3. In column (3) we analyze attrition from round 3 to round 4 by using round 3 characteristics and a dummy for if the household is still in the sample in round 4. In column 4 we compare households that left the sample in round 1, 2, or 3, with households that remained in the sample in every round.

While in some rounds, households with certain characteristics were more likely to drop out of the sample, in our analysis we focus on consistent predictors across the three rounds. Overall, households with a female survey respondent were less likely to leave the sample. Respondents with higher education were more likely to drop out of the sample. In terms of household characteristics, household size, number of dependents, or whether the household had only female members, were not significant predictors of attrition. Further, households who moved more recently were not more likely to drop out of the sample. While having a household of improved build was not associated with attrition by round, attrition households were less likely to have an improved house than households that remained in the sample in every round.

**Table 5. Overview of how often households were interviewed across the four survey rounds, by survey round.**

| *# times that sample households were interviewed among the four rounds*: | Round 1 | | Round 2 | | Round 3 | | Round 4 | |
|---|---|---|---|---|---|---|---|---|
| | **N** | **%** | **N** | **%** | **N** | **%** | **N** | **%** |
| One | 3,725 | 30.8% | 1,280 | 10.5% | 828 | 6.8% | 3,724 | 28.8% |
| Two | 2,043 | 16.9% | 2,513 | 20.7% | 2,610 | 21.5% | 1,670 | 12.9% |
| Three | 1,501 | 12.4% | 3,518 | 29.0% | 3,859 | 31.8% | 2,699 | 20.9% |
| Four | 4,831 | 39.9% | 4,831 | 39.8% | 4,831 | 39.8% | 4,831 | 37.4% |
| *Total # households interviewed* | *12,100* | | *12,142* | | *12,128* | | *12,924* | |

Source: Authors' estimates from 2022 MHWS

**Table 6. Marginal effects of probit regression analyses of attrition (1–4) among all survey respondents, and among replacement and attrition respondents only (5–6).**

| | Attrition (all survey round respondents) | | | | Attrition (among replacement and attrition households only) | |
|---|---|---|---|---|---|---|
| | R1 (1) | R2 (2) | R3 (3) | R1, R2, R3 (4) | R2 (5) | R3 (6) |
| Respondent's age (#) | -0.00*** | -0.00*** | -0.00*** | -0.01*** | 0.00*** | 0.00 |
| Respondent is female (0/1) | -0.01 | -0.02** | -0.03*** | -0.04*** | -0.02 | -0.01 |
| Respondent has at most primary education (0/1) | 0.03*** | -0.01 | 0.02** | 0.02** | 0.02 | -0.09*** |
| Dependents (#) | 0.00 | 0.00 | -0.01 | 0.00 | 0.02** | 0.01 |
| Household size (#) | 0.01** | 0.00 | 0.00 | 0.01*** | 0.02*** | 0.00 |
| Women only household (0/1) | 0.01 | -0.03 | 0.00 | 0.01 | -0.02 | 0.00 |
| Moved to current village < = 2 years ago (0/1) | 0.01 | 0.03 | 0.02 | 0.01 | 0.14*** | 0.10*** |
| Improved house [a] (0/1) | -0.02 | -0.01 | -0.01 | -0.02*** | -0.03 | 0.03 |
| Household has electricity connection [b] (0/1) | -0.04*** | -0.02* | 0.00 | -0.03*** | 0.01 | 0.00 |
| Agriculture land > 0–2 acre (vs no land) | 0.02 | -0.04*** | -0.01 | 0.00 | -0.04 | 0.03 |
| Agriculture land 2–4 acre (vs no land) | 0.01 | -0.03** | 0.01 | 0.00 | -0.03 | 0.06** |
| Agriculture land 4–7.5 acre (vs no land) | 0.02 | -0.05*** | 0.02 | -0.01 | -0.06** | 0.09*** |
| Agriculture land >7.5 acre (vs no land) | -0.01 | -0.03** | 0.01 | -0.03** | 0.01 | 0.11*** |
| Farm wage (vs own farm) | 0.00 | 0.01 | 0.00 | 0.01 | 0.05* | -0.05* |
| Non-farm wage (vs own farm) | -0.03* | -0.01 | 0.01 | -0.01 | 0.01 | -0.03 |
| Non-farm salary (vs own farm) | -0.03* | 0.00 | -0.02 | -0.02* | -0.01 | -0.03 |
| Non-farm business (vs own farm) | 0.00 | 0.00 | 0.00 | -0.01 | 0.05** | 0.02 |
| Other income vs own farm | -0.02 | 0.00 | 0.01 | -0.01 | 0.04 | -0.09** |
| Income quintile 2 vs (income quintile 1) | 0.02 | -0.02* | -0.03** | -0.01 | 0.02 | -0.05** |
| Income quintile 3 vs (income quintile 1) | 0.01 | -0.04** | -0.02 | -0.03** | 0.03 | 0.01 |
| Income quintile 4 vs (income quintile 1) | 0.00 | -0.01 | -0.02* | -0.03** | 0.10*** | 0.02 |
| Income quintile 5 vs (income quintile 1) | -0.01 | -0.02* | -0.01 | -0.02** | 0.09*** | 0.06** |
| Received remittances (0/1) | -0.02 | -0.01 | 0.02* | 0.00 | 0.00 | 0.05** |
| Battles/explosions/remote violence (#) | 0.00 | 0.00 | 0.00 | 0.00 | 0.00** | -0.01 |
| Feels insecure in community (0/1) | -0.02* | -0.03*** | -0.02* | -0.03*** | -0.03 | 0.00 |
| Low level of social trust in community (0/1) | -0.01 | -0.02 | 0.00 | -0.02* | 0.01 | 0.01 |
| Climatic shock (0/1) | -0.02 | -0.01 | 0 | 0.01 | -0.08*** | -0.12*** |
| Altitude (metres) | 0.00** | 0 | 0.00*** | 0.00*** | 0.00*** | 0 |
| Remote (0/1) | 0.03*** | 0.03*** | 0.01 | 0.03*** | 0.01 | 0 |
| Rural (0/1) | -0.02 | 0 | -0.02 | -0.02** | 0.02 | -0.05** |
| No. of Obs. | 11514 | 11748 | 11881 | 23942 | 4714 | 3806 |

[a] Improved house refers to houses made from semi-pucca, bungalow/ brick, apartment/ condominium.

[b] Electricity connection includes households connected by national grid/private company/generator

Notes: Stars denote significant differences at p-values

* p < 0.10,

** p < 0.05,

*** p < 0.01.

Source: Authors' estimates from 2022 MHWS

Finally, households with an electricity connection were less likely to attrite in most rounds and compared to panel households in all three rounds.

For the most part, there was no association between attrition households and their main source of income. The only exception was that compared to farm households, households whose main source of income was salaried work were less likely to drop out of the sample

between round 1 and round 2 and compared to the households that remained in the sample in every round. While between round 1 and round 2, compared to households in the lowest income quintile, households in higher income quintiles were not more likely to leave the sample, in round 2 and round 3, wealthier households do appear more likely to attrite, though the relationship does not appear to be linear. On the other hand, compared to households that remained in the sample in every round, households in the three highest income quintiles were less likely to attrite. Importantly, shocks appear to have no impact on attrition. In fact, households who reported feeling insecure were less likely to attrite in every round. Finally, remote households were more likely to attrite across the rounds. Overall, it appears that wealthier households were more likely to remain in the sample, but only by a very marginal amount.

We also investigate the differences between replacement households and attrition households in columns (5–6). In column (5) we analyze attrition from round 2 to round 3 by using round 2 characteristics. The sample therefore consists of round 2 households who either newly entered the sample in round 2 (i.e., a replacement household), or who we know will drop out in round 3. In column (6) we similarly analyze attrition from round 3 to round 4 by using round 3 characteristics, and we define a replacement household as a household who joins the sample in round 3 and was not present in round 2.

Compared to replacement respondents, between round 3 and round 4, attrition households were less likely to be low educated. Attrition households were significantly more likely to have moved in the past two years compared to replacement households. This suggests that we struggled to add displaced households to our survey, despite the increase in displacement in Myanmar over the year. There were not clear patterns between attrition and replacement households in terms of agricultural land holdings and source of income. Attrition households were more likely to be in the wealthiest income quintile, compared to the poorest. Finally, attrition households were less likely to experience a climatic shock, compared to replacement households. Overall, however, the effect sizes of these differences were small, which suggests that our sampling strategy successfully replaced attrition households with households with similar observable characteristics.

## Section 4: Discussion

In this paper, we described an approach to implementing a nationally and sub nationally representative phone survey starting from a non-representative database of phone numbers. The ingredients in this approach were:

1. A large and geographically dispersed database of phone numbers, in this case independently generated by the collaborating survey firm;

2. A quota-based sampling strategy designed to reduce common phone survey biases (such as geographical bias, over-sampling of more educated and urban respondents) and to achieve gender parity as well as an over-sampling of sub-samples of interest (in this case, farm households); and

3. A multi-step construction of survey weights designed to further improve national and subnational representativeness.

Overall, the approach outlined in this study appears to be remarkably successful in generating a new nationally and regionally representative phone survey data set (i.e. the MHWS). Building a large database of phone numbers to work from (as in our case) is likely a major advantage and is necessary to conduct representative phone surveys. One way to do this is to start with RDD (manually or by using a system) along with continuous snowball sampling

recruitment. Another way is hiring local recruiters. Or instead, one could build on the most recent nationally representative in-person survey and call back the participating households with at least one phone number listed [25].

In terms of geographical coverage, the quota-based approach was relatively effective in reducing bias towards respondents from more geographically accessible townships or towards urban-based respondents. Compared to formerly conducted nationally representative datasets in Myanmar it covers more townships than any of its predecessors. All existing nationwide face-to-face survey efforts in Myanmar have been significantly hampered by inaccessibility, insecurity, and travel restrictions. Past in-person survey teams had to avoid entire townships affected by conflict, but our phone survey sample does not contain such geographic bias. An estimated 1.2 million people (about 2.3 percent of the total population) could not be enumerated during the 2014 Census [15]. The 2019 ICS excluded eight townships in Shan State from the ICS sample frame due to expected inaccessibility, but still failed to enumerate 8 percent of the remaining ICS enumeration areas [22]. In Rakhine State, the State which experienced the largest number of conflict and violent events at that time, the 2019 ICS was unable to reach about 74 percent of the selected enumeration areas [22]. These figures clearly demonstrate the challenges of implementing face-to-face surveys in Myanmar, even at a time when conflict was less widespread compared to in 2022. Biases towards reaching more respondents in well-connected and urban areas were also noted during phone survey efforts in Ghana, Nigeria, and Burkina Faso [6, 9, 12].

The MHWS also performs well in terms of household-level representativeness when considering several key socio-economic indicators compared to other phone survey datasets that have been well documented and thoroughly vetted [5, 7, 10]. Often household weights may only partially reduce such bias [9]. However, in our survey, weighted statistics for key variables that are roughly time-invariant closely match findings from other recent nationally representative surveys.

More remarkable though is the reasonable approximation of individual respondent characteristics to the general adult population in terms of education level, age, the relation to the household head and gender. Other studies find that their respondents are generally younger, better educated, and either include disproportionately more male or more female respondents compared to the average population in their study area [2, 6, 9, 12]. Part of the success of MHWS in being more representative of the adult population is a result of its quota-based sampling strategy and its purposive decision to interview any adult respondent that answers the phone, regardless of his or her role in the household. This approach contrasts with many data collection efforts which often intentionally aim to interview the household head or the most knowledgeable respondent related to the topic of interest, or have no choice but to call the household head as this is the only phone number recorded during former in-person data collection [7, 10]. At the same time, by doing this we might be trading off accuracy of data about the household at large for better accuracy of data related to the individual respondents themselves. More research is needed to understand if and how accuracy of household data changes when any household member is interviewed rather than the most knowledgeable member.

Inevitably, there is some residual bias–observed and unobserved—in the survey sample, even after corrections using survey weights. Older people are underrepresented in our sample–both prior to and after weighing–compared to the age distribution of adults in the regular population, likely because of lower phone ownership among this group. Further, our assessment is limited to the available indicators comparable between our survey and other benchmark surveys, and one might expect further bias in certain respondent characteristics that we could not measure or control for. Geographically, even if at the township-level we have an adequate distribution, it is likely that residents of more remote locations or of acutely conflict-

affected areas within each of these are not or less reached. In terms of demographic and wealth characteristics, we likely underrepresent respondents who do not easily communicate in the main language (Burmese) or one of the other major ethnic languages and those at the highest and lowest percentage of the wealth distribution–the latter are likely non-phone owners. Many of these concerns, however, also pertain to in-person survey efforts.

Whether the MHWS out- or underperforms in-person surveys in terms of attrition rates is difficult to assess given that we ideally would compare it to a survey conducted at a similar frequency (the interval between MHWS survey rounds was only three months), and in a similar setting (heavily impacted by severe shocks). Moreover, attrition can be further mitigated by devoting more effort to tracking respondents. In our survey, we only ask enumerators to call back the same household five times before replacing the household. In-person interviews are less commonly done in settings that are heavily impacted by severe shocks, and in-person panel surveys are often (though not always) implemented several years apart. In Myanmar's case, no in-person nationwide panel household surveys were conducted in the past. The Integrated Household Living Conditions Assessment (IHLCA) survey conducted in 2004 and 2009 was only a half-panel, i.e., only half of all households in each survey cluster were re-interviewed whereas the other half of interviewed households newly entered the sample [26].

An interesting comparison is a recent study in Malawi and Liberia, where respondents were given cell phones and were administered a phone survey every two months. The surveys were initiated prior to, but continued during, the COVID-19 pandemic [27, 28]. Attrition between rounds was lower in Malawi compared to Liberia, likely because of a better cellular network in the former. In Liberia, attrition averaged 49 percent in 2021 and increased over time, as respondents changed phone numbers or opted out of the surveys. When phone surveys were conducted in Ethiopia during conflict, respondents from the severely affected Tigray region were not interviewed at all (25 percent of the sample), but attrition rates among respondents in the other regions were low (1 percent of the remaining sample) [29]. When comparing the response rate to the COVID-19 high frequency surveys implemented among participants in recent in-person Living Standard Measurement Surveys (LSMS) in five different countries in Sub-Saharan Africa, as many as 94 percent of households with phone numbers were reached in Uganda but only 62 percent Ethiopia [10].

MHWS also has some notable advantages over face-to-face surveys when considering potential time-savings in conducting such a large-scale survey effort. The survey was designed and deployed in a very short time span (a few months), and at a high frequency—four times in 2022. This is particularly advantageous in volatile economic and political settings. Further, face-to-face surveys, such as MLCS, use a two-stage survey design and cluster typically between 12 and 30 survey households within each enumeration area to reduce transport and other logistical costs. Clustering strategies used in in-person surveys can reduce precision of indicator estimates, but such an approach is not necessary for phone survey interviews.

Phone surveys also have other limitations, for example related to the interview duration. Phone surveys are short-duration interviews with a limited number of questions relative to in-person interviews. The average time for an MHWS questionnaire by phone was 36 minutes, whereas in-person surveys often last three to four times longer. The in-person MLCS, for example, had an average interview duration of 140 minutes [21]. A setup of rotating modules that are asked only once across different phone survey rounds could try to attenuate such shortcoming, though even across four survey rounds a phone survey is unlikely to capture as much as an in-person survey.

Studies have also increasingly reported concerns over data quality. There is some evidence that lower enumerator trust and higher measurement error in phone surveys means that responses from phone surveys can be systematically different from or less precise than those of

in-person surveys [12, 29–31], and that respondent fatigue may be at least as problematic in shorter phone surveys as it is in longer in-person surveys [32, 33] and may also occur in repeated surveys [33]. Also, recent work showed that who responds to the survey matters [34], thus researchers may need to make a conscious decision on whether to interview any or rather a specific household member to achieve the most accurate estimates of their key indicators of interest. That said, more research is needed to assess whether these are widespread problems or particular to the studied populations, survey modules, and modalities.

## Section 5: Conclusion

While the use of cell phone surveys in lower-middle income countries has increased over the last decade, COVID-19 led to a jump in cell phone surveys, but only a few of these aimed for and claimed national representativeness–let alone sub national level representativeness. A healthy dose of caution is indeed warranted in assuming that "national" phone surveys are representative [7]. Phone ownership is not universal, and some respondents may be more able and willing to respond to a phone survey than others, thus potentially leading to over- or underrepresentation of certain groups among phone survey respondents. National in-person household survey efforts, however, also encounter households refusing to be interviewed and they often require replacement of randomly selected enumeration areas with a nearby area due to inaccessibility for safety or other reasons. In some fragile states, such as Myanmar, entire geographical areas have been avoided by survey teams, thus leading to a non-negligible part of the population not being enumerated at all.

We demonstrated an effective method for designing and implementing phone surveys that are nationally and sub nationally representative in areas affected by natural or human-made shocks. In doing so, our example from Myanmar serves to encourage researchers to consider phone survey data collection as a different yet effective alternative to in-person surveys. In several contexts, the ability to reach people in areas affected by natural or human-made shocks may outweigh potential disadvantages of phone surveys related to a shorter interview duration, the lack of in-person interactions, data quality or others.

Collecting nationally and sub nationally representative data on key welfare indicators (within the range of what is feasible within the short duration of a phone survey) is critically important in fragile states such as Myanmar, where reliable data and rigorous research are increasingly scarce, yet also vitally important for targeting more resources to a growing population of vulnerable people. Monitoring individual and household welfare on a more frequent basis, however, is important in almost any LMIC context. Previous research already advocated for high frequency surveys focusing on the importance of climate change and other ecological and economic transformations, as well as seasonal shocks [35, 36]. Agricultural economies are volatile at the best of times, but even urban economies in less developed countries are clearly highly vulnerable to the threats of further pandemics [37] and are affected by more frequent severe weather events induced by climate change [38]. Phone surveys can gauge many of the key impacts of these shocks and identify vulnerable households, the effectiveness of their coping mechanisms as well as external interventions, and potentially identify key trends–especially in agriculture–to inform early warning systems.

## Supporting information

**S1 File.**
(DOCX)

## Author Contributions

**Conceptualization:** Bart Minten, Hnin Ei Win.

**Data curation:** Isabel Lambrecht, Joanna van Asselt, Patrick Meza, Moe Sabai, Thet Su Sun.

**Methodology:** Joanna van Asselt, Bart Minten, Hnin Ei Win.

**Project administration:** Bart Minten.

**Visualization:** Moe Sabai.

**Writing – original draft:** Isabel Lambrecht.

**Writing – review & editing:** Joanna van Asselt, Derek Headey.

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
