## [Decision Letter · Decision Letter 0]

5 Sep 2023

PONE-D-23-25141Phone surveillance – from scratch: National and subnational level data collection in MyanmarPLOS ONE

Dear Dr. Van Asselt,

Thank you for submitting your manuscript to PLOS ONE. After careful consideration, we feel that it has merit but does not fully meet PLOS ONE’s publication criteria as it currently stands. Therefore, we invite you to submit a revised version of the manuscript that addresses the points raised during the review process.

I think the reviewers pointed out some very important points, and I hope that the authors will take their comments into account to improve this manuscript.

We look forward to receiving your revised manuscript.

Kind regards,

Seo Ah Hong, PhD

Academic Editor

PLOS ONE

[The authors do not have any competing interest regarding the publication of the manuscript. Publication of the manuscript will not affect funding, employment or other financial and non-financial interests of the authors, neither from IFPRI-affiliated nor from Myanmar Survey Research – affiliated authors]. 

6. We note that Figure 1, S1, S2 and S3 in your submission contain [map/satellite] images which may be copyrighted. All PLOS content is published under the Creative Commons Attribution License (CC BY 4.0), which means that the manuscript, images, and Supporting Information files will be freely available online, and any third party is permitted to access, download, copy, distribute, and use these materials in any way, even commercially, with proper attribution. For these reasons, we cannot publish previously copyrighted maps or satellite images created using proprietary data, such as Google software (Google Maps, Street View, and Earth). For more information, see our copyright guidelines: http://journals.plos.org/plosone/s/licenses-and-copyright.

A. You may seek permission from the original copyright holder of Figure 1, S1, S2 and S3 to publish the content specifically under the CC BY 4.0 license.  

B. If you are unable to obtain permission from the original copyright holder to publish these figures under the CC BY 4.0 license or if the copyright holder’s requirements are incompatible with the CC BY 4.0 license, please either i) remove the figure or ii) supply a replacement figure that complies with the CC BY 4.0 license. Please check copyright information on all replacement figures and update the figure caption with source information. If applicable, please specify in the figure caption text when a figure is similar but not identical to the original image and is therefore for illustrative purposes only.

8.We notice that your supplementary figures are uploaded with the file type 'Figure'. Please amend the file type to 'Supporting Information'. Please ensure that each Supporting Information file has a legend listed in the manuscript after the references list.

Reviewers' comments:

Reviewer's Responses to Questions

**Comments to the Author**

1. Is the manuscript technically sound, and do the data support the conclusions?

Reviewer #1: Yes

Reviewer #2: Yes

Reviewer #3: Partly

2. Has the statistical analysis been performed appropriately and rigorously? 

Reviewer #1: Yes

Reviewer #2: Yes

Reviewer #3: Yes

3. Have the authors made all data underlying the findings in their manuscript fully available?

Reviewer #1: Yes

Reviewer #2: Yes

Reviewer #3: No

4. Is the manuscript presented in an intelligible fashion and written in standard English?

Reviewer #1: Yes

Reviewer #2: Yes

Reviewer #3: Yes

5. Review Comments to the Author

Reviewer #1: this manuscript presents an approach to a large (n=12k) panel cell phone survey in myanmar. authors details the sampling and weighting approach, then present multiple sections of results to show the comparability of their phone survey with recent FTF surveys.

this paper is important as the cell phone survey field continues to grow.

i detail to authors areas for clarification and updates in an attachment to this review.

the results section can be half the current length. there is too much detail ; please make the results more succinct.

more information about methods is needed and requested in the attachment.

a few times authors use terms interchangeably, or inconsistently, but overall the manuscript is well written.

i look forward to seeing revisions. (please see additional comments in the attached file.)

Reviewer #2: The paper is excellent in using rigorous methods to assess the representativeness of a phone survey sample. I also commend the authors for being diligent in documenting the methods used and the transparency. This is one of the few studies that I have seen that have been very rigorous and in diligently documenting methods used for assessing representativeness and in weighting. (Please see additional comments in the attached file.)

Reviewer #3: Thank you for the opportunity to review this paper.

The paper ‘Phone surveillance – from scratch: National and subnational level data collection in Myanmar’ discusses the experience of designing and implementing a phone survey in Myanmar – from scratch, that is, not building on a previous in-person survey. The paper makes a meaningful contribution and in principle deserves publication in PLOSONE, which has previously published other important studies on this issue. However, the paper is not fit for publication in its current form and needs to be revised considerably in several respects.

I divide my comments into major and minor comments.

Major comments:

1. Non-sampling errors in phone surveys. The paper argues that phone surveys should not be considered second-best to in-person surveys suggesting that the phone survey in question outperformed previous in-person surveys. But the paper focuses on sampling design, showing the success of this phone survey in that domain. However, a key advantage (arguably the most important one) of in person surveys is that they allow for longer interviews, with more intensive interviewer supervision. As a result, there seem to be limitations to what data can be collected over the phone and questions about how reliable the data is, i.e. phone surveys may be greater non-sampling error. I see no serious discussion of this aspect. Conclusions about the utility of phone surveys relative to in-person will depend critically on this aspect. Relatedly, it would be good to understand what data items were collected in this phone survey and how reliable we can deem the data. E.g. it appears ‘poverty’ was monitored. Does this mean the survey collected a full consumption module over the phone? How successful was this? Etc. This also speaks to the question if and how phone surveys can be used for welfare monitoring.

1. Attrition: The attrition rates in this survey are severe and so attrition deserves to be discussed in more detail. How does it compare to other phone surveys? How does it compare to in-person surveys? The extent of attrition relative to other types of phone or in-person surveys is a key element in determining which survey mode can be considered reliable or first-best. In light of this, it may not be warranted to conclude that the phone survey in question performed better than other phone surveys – it certainly seems to have succeeded in reaching the targeted respondents but its success also depends on such factors as attrition and non-sampling errors. Relatedly, it would be good to show more clearly the attrition rates e.g. in a table.

2. Broader policy implications: The paper summarizes the design of a successful phone survey in Myanmar. What are the policy implications and relevance for contexts beyond this specific one? This specific survey relied on a large database from which the survey designers could build their sample – is a phone survey from scratch only possible with a database like this? It is not something survey designers and researchers always have at their proposal. So how does this get replicated? Do NSOs or survey firms start putting together a database? Should this be supported by INGOs/IOs etc.?

3. Sampling of respondents: The lack of representativeness in other phone surveys (e.g. those discussed in the Brubaker et al. paper that is cited frequently here) is due to the deliberate targeting of a knowledgeable adult in those surveys who can respond on behalf of the entire household. The approach followed in this survey is a convenience sampling approach. Two notes: first, there may be a trade-off here between choosing a reliable respondent and achieving better representativeness. Second, from a sampling standpoint, best practice is a probability selection method rather than selection by convenience. It may be worth discussing these points.

4. Discussion of weighting methods. These sections should be more detailed and may benefit from a more formal treatment to make it clearer to readers how these methods work.

Small comments

-Comparison to in-person surveys: This paper relies on the number of townships reached (or not reached) as a metric of their success. Is this appropriate? A survey may not need to reach all townships to be representative. Specifically, it would be good to understand if the in-person surveys excluded certain townships because those were not selected due to sampling design or because those should have been selected but could not be reached due to insecurity.

-Introduction: Phone surveys were limited in low-income contexts but very common in Europe and North America for instance.

-Incentives: Clarify if any incentives were used

-S6 Table shows that differences between adult respondents in the MHWS and the adult population in the MLCS further reduce after weighting the MHWS data rather than aggravate as in Brubaker et al. (2021).” It is my reading of the Brubaker et al paper that they find that sampling weights reduce rather than aggravate differences, but do not reduce enough to overcome significant differences.

6. PLOS authors have the option to publish the peer review history of their article (what does this mean?). If published, this will include your full peer review and any attached files.

Reviewer #1: No

Reviewer #2: No

Reviewer #3: No

---

## [Author Response · Author response to Decision Letter 0]

24 Oct 2023

October 20, 2023 

Ref: Resubmission of Manuscript PONE-D-23-25141

Dear Editor:

We appreciate the opportunity to revise our original research manuscript "Phone surveillance – from scratch: National and subnational level data collection in Myanmar.” We revised the document according to your recommendations and the reviewers’ thoughtful comments. The most noteworthy changes are:

As suggested by reviewer 3, we toned down the statements that argue that a phone survey is equal to an in-person survey. The intended focus of our paper was on demonstrating a sampling and weighting method to make a phone survey nationally and sub nationally representative, not on proving that a phone survey may be as good as an in-person survey. We have removed all the language that indicates that a phone survey may not be second-best compared to an in-person survey. Further, we better highlight the weaknesses of phone surveys including shorter survey duration, lower enumerator trust, and possible measurement error. 

In addition, and to satisfy reviewer 1’s request, we have shortened the discussion in the results section and either moved this discussion to the conclusion or dropped it all together. Further, we correct for all the inconsistencies or clarify confusing statements that reviewer 1 diligently pointed out. 

Finally, we have improved our discussion of attrition. We have changed our main table on attrition in the text, as well as added two supplementary tables to emphasize when households left and re-entered the panel. We also try to make comparisons to attrition rates in other phone surveys.

Please refer to following pages for our detailed response to each of the reviewers’ comments. 

Thank you for your consideration of this manuscript. We look forward to your reply.

 

Reviewer 1 

Thank you very much for your careful reading and thoughtful assessment of our initial 

manuscript. In this revised version, we have endeavored to implement your suggestions and 

address your concerns, as well as those of the other reviewer and of the editor. Below we explain 

in detail how we have handled your specific comments. For clarity, we first reproduce your 

original comments, in bold italics, followed by our response, in roman font, and the new lines in the text in italics. 

 in LMIC. in US or europe phone and sometimes web are widely accepted

Thank you for your comment, this is an important point, we have updated the first line of our abstract, as well as the title of the paper, page 1, paragraph 1 of the abstract: 

For decades, in-person data collection has been the standard modality for nationally and sub-nationally representative socio-economic survey data in lower middle-income countries. 

 what does "from scratch" mean? and it seems you're using an existing list of phone number 

Again, thank you for your comment. We deleted from scratch from this line, as well as throughout the paper, and in the title. The sentence you refer to in the text, page 1, paragraph 1 of the abstract, now reads as follows:

Our study proves the feasibility of collecting nationally and sub nationally representative phone survey data, which is critical for rapid monitoring in any volatile economy. 

 wealthier would be more formal language

Thank you for pointing this out, we have changed the word to wealthier, page 2, paragraph 3 of section 1, as suggested. 

Moreover, mobile phone owners are typically wealthier, more educated, younger, and more male than non-owners (L’Engle et al. 2018; Lau et al. 2019; Gourlay et al. 2021).

 this article that talks about how bias is more important than response rate might be a nice citation to include:

We thank the reviewer for sharing the citation. We have read the article and added the citation. 

Poorer people, farmers, or women may be less likely to answer a phone and could therefore be under-represented when randomly calling phone numbers (Johnson and Wislar 2012; Gourlay et al. 2021). 

 a list ie a sample frame , when looking at the total survey error framework, sample frame comes before non-response so I might switch the order of these paragraphs 

The order of the paragraphs has been switched. Further, the term sampling frame has been added to page 2, paragraph 2 of section 1, to clarify what we mean by comprehensive list. 

One challenge when setting up a phone survey is getting access to a sampling frame, which for a phone survey is a comprehensive list of phone numbers.

 citation for this?

In the absence of any other feasible or affordable alternatives and considering the

urgency of data collection needs during the early stages of the COVID-19 pandemic, many phone surveys relied on contact information from existing non-representative program-based surveys. 

Thank you for your comment, in the end we decided to delete the sentence, because these surveys did not intend to be nationally representative in the first place, which was our goal. 

 do you have a citation for this or is it just conjecture that RDD is more expensive? there are ways to reduce # of calls with RDD by validating phone #s, etc.

Thank you for your comment, we have added a citation in page 2, paragraph 4 of section 1. 

Random digit dialing or calling respondents from telecom provider lists likely necessitates large amounts of phone calls to find respondents from these specific subpopulations - especially if regions are diverse in population size, phone ownership or response rates - thereby raising the cost of the survey quite substantially (Dillon et al. 2021).

 a segment of the pop that has low mobile phone ownership is going to be excluded regardless of where the sample frame came from. if you're using an existing list of phone numbers, if an areas has low mobile phone ownership, it's still going to be hard to get ahold of ppl 

Thank you for pointing out this inconsistency. The reviewer is correct, the point we were trying to make is that because in-person surveys struggle to reach people from difficult geographies, the combination of that and low mobile phone ownership in those areas may mean a very small sample to draw on. We have clarified the sentence as follows on page 2, paragraph 4 of section 1:

Using contact information from in-person survey datasets may result in underrepresentation of certain geographies or segments of the population who were difficult to travel to in-person.

 greenleaf, plosOne; burkina faso 2019 article

Thank you for drawing our attention to this paper. We have read the paper and added this citation in the relevant lines of the paper. 

Further, the use of sampling weights cannot always correct these biases to ensure representativeness – especially at the individual level (Brubaker et al. 2021, L’Engle et al. 2018; Lau et al. 2019; Greenleaf et al. 2020).

 this paragraph seems out of flow

Thank you. We have deleted the paragraph to make the writing more condensed and consistent. 

 interest in phone surveys in lmic increased precipitously in ~2015 then was widely used during COVID. so ppl already know they're useful beyond a pandemic.

Thank you for your comment. We now try to highlight that the use of phone surveys was already increasing in LMICS prior to COVID-19 in the beginning of the introduction. Further, in response to this comment as well as another of your comments we ended up deleting the sentence you refer to, and changing the language in the remaining sentence on page 3, paragraph 2 as follows: 

We demonstrate how phone surveys can be a powerful tool to collect socioeconomic data in fragile states.

 you're not starting from scratch - informal term and you have a sample frame of phone numbers

Thank you for your comment. Yes, you are right, we have corrected this throughout the paper and in the title. Please see the new sentence on page 3, paragraph 5 of section 1, which your comment directly addressed. 

Second, the paper provides details on an innovative sample design and weighting strategy to minimize bias in characteristics typically associated with phone surveys, even when starting with a non-representative sample frame. 

 this paragraph and previous are too long. just state the objectives of the paper. you have already set-up the rationale

We have cut sentences from both paragraphs to condense the two paragraphs into one paragraph on page 3, paragraph 5 of section 1, that is more focused on the research objectives of the paper. 

 really important to share the % of the population that owns a cell phone in myanmar. that will help contextualize your results to others - my guess is ownership is high which is why your weights work better than in other geographies where weights haven't helped 

We have added estimated from the 2019 Inter-Censal Surve on page 5, paragraph 1 of section 2.1 The MHWS sample design.

The 2019 Inter-Censal Survey (ICS) estimates that in 2019, 85.8 percent of households owned mobile phones, 82.2 percent of rural households, and 94.9 percent of urban households. At that time, Ayeyawaddy region had the lowest mobile phone ownership, 76.0 percent of households, while Yangon had the highest, 94.4 percent of households (16).

 I don’t think 4 calls makes a survey high frequency - and across what amount of time? how are you defining high frequency? this is more like repeated cross-sectional you did phone calls quarterly - i dont consider that high freq but maybe you have a different definition?

Thank you for your comment, we agree with your point and have changed our language to a quarterly phone survey. We do not use the term repeated cross-sectional because we have an unbalanced panel. We have made these changes throughout the paper to replace the term high frequency. Please find below the specific example you refer to in your comment on page 3, paragraph 7 of section 1. 

We therefore organized a large-scale quarterly phone survey, representative at the national level, the urban/rural level, and the state/region level, called the Myanmar Household Welfare Survey (MHWS). 

 did you ask the same questions each round? did interviewers administer the survey? 

Thank you for your comment. We have added a few sentences to clarify this point. 

Page 4, paragraph 1 of section 2. 

In each round, the same questions were asked to capture changes in the indicators over time.

Page 4, paragraph 2 of section 2. 

Interviews were conducted by enumerators using the KoboCollect App, a software used for data collection that is compatible with Computer Assisted Telephone Interviews (CATI) software. Enumerators conducted the interviews at the MSR office. 

 can we get # of states and # of regions? How many?

We have added the number of states/regions in the text, page 5, paragraph 2 of section 2.1, as well as we have added a supplementary figure that shows the states/regions with the following clarifying note below the figure. 

The MHWS intended to interview 12,790 respondents for the 15 States and Regions in Myanmar (S1 Fig).

There are seven regions; they are areas that are predominately Burman (Bamar). There are seven states, they are home to Myanmar’s most politically and numerically dominant ethnic minorities. There is one union territory, Nay Pyi Taw, which we include as a state/region.

 do you happen to know over what period this database was created and were these participants enrolled via RDD or from another existing sample frame?

We have added the following clarifying sentence to explain the time frame and the methods that MSR used to create the database, page 5, paragraph 2 of section 2.1.

The MSR database was first created in 2014. It was established through a combination of random digit dialing and referrals. Furthermore, starting in 2015, after concluding several opinion polls and market research studies, respondents were asked to provide consent for future survey work.

 you need to mention in the abstract that you use an existing sample frame of phone #s for the survey. this is important to share and is more specific than saying "from scratch"

Thank you for your comment. We have added the following sentence to the abstract. 

Our study proves the feasibility of collecting nationally and sub nationally representative phone survey data using a non-representative sample frame, which is critical for rapid monitoring in any volatile economy.

 ie sample frame in survey methods terms

We have added the term sampling frame to use the correct terminology, page 6, paragraph 3 of section 2.1. 

The first step in selecting phone numbers for interview was the development of a sampling frame or master phone number database.

 you did PPS at state, region and township level? this paragraph and the first paragraph are a bit confusing together

Thank you for your comment. We have clarified the paragraphs on page 6, paragraph 3 of section 2.1 as follows: 

To create the master database all phone numbers were stratified at the township level, which is a smaller geographical unit within each state or region, numbering 330 in total. The final amount of phone numbers in the master database were proportional to the population size in each township, based on the information in the 2014 Census data, but contained four times the actual number of target interviews for each township (to account for non-response).

 how did you identify these townships?

The township proportionality structure of this master database was designed to minimize the risk of oversampling respondents who live predominantly in well-connected and wealthier townships.

Face-to-face surveys, such as MLCS, use a two-stage survey design, and cluster typically between 12 and 30 survey households within each enumeration area to reduce transport and other logistical costs. But this structure can oversample respondents in well-connected wealthier townships because they are easier and cheaper to reach. Instead of use two-stage survey design, we tried to enumerate every township relative to the population size of the township, and in that way tried to avoid oversampling wealthier more well-connected townships. 

 when you say sample, do you mean the sample frame or the actual sample that was enrolled?

We intended to say the final sample of interviewed respondents, and we have clarified that in the sentence on page 6, paragraph 3 of section 2.1: 

While we did not insist on having an exact proportional balance of interviews at township level in the final sample of interviewed respondents, the survey company did strive to achieve such balance to the extent possible.

 this sounds like a quota - why aren't you using the term quota? 

You are right, we have changed the term to quota for consistency, in the line you refer to here as well as throughout the paper. On page 6, paragraph 4 of section 2.1:

We therefore set the following quotas at State/Region level (S2 Table):

 you should write in the abstract as well that you used quota sampling. saying quota is more succinct that what you have and is v informative 

We have added the following sentence to the abstract to clarify that we use a quota sampling technique. 

We use quota sampling and entropy weighting to obtain a better geographical distribution compared to recent in-person survey efforts, including reaching respondents in areas of active conflict.

 did interviewers work in a call center? how many of them? did you use any software? answers should go in methods section? 

The following sentences have been added to the methods section to clarify this point, page 4, paragraph 2 of section 2:

Interviews were conducted by enumerators using the KoboCollect App, a software used for data collection that is compatible with Computer Assisted Telephone Interviews (CATI) software. Enumerators carry out the interviews at the MSR office. Only when given permission by the manager do enumerators make calls from home, but this is only for appointments after working hours. 

 Do you mean if the person who picked up the phone screened into an age group that had already reached saturation, the interviewer sought another household member who may fit into the desired age range?

Sorry for our initial lack of clarity. We have clarified the sentence as follows on page 7, paragraph 6 of section 2.1: 

If the initial respondent who picked up the phone was not eligible to be interviewed because they were either too young or too old, the interviewer asked to speak with another household member whose age was in the desired age range.

 what do you mean by 'target get'? your quotas? you didn't meet your quotas in certain states?

We have edited the language in this entire paragraph to highlight the two issues, not reaching our target sample size and not reaching some of the quotas. 

 not understanding this second to last sentence - how was the collection altered? before what were you doing? and what did you then do?

Thank you for your comment and sorry for our lack of clarity. We tried to clarify this point by editing the paragraph on page 8, paragraph 8 of section 2.1: 

From the beginning, enumerators were required to call a respondent five times at different times of day before dropping the household from the sample. In rounds 1, 2, and 3, we did not offer explicit descriptions of the amount of time to wait between phone calls, and 2) the number of respondents called in each state/region. In round 4, we required enumerators to 1) wait at least 3 days between phone calls to the same respondent, and 2) stratify calls to each state/region over the three-months of data collection, with 25 percent of phone calls per state/region per month.

 just say 'population' not 'in terms of the sample'

We have deleted the term 'in terms of the sample'. The updated sentence below, page 9, paragraph 1 of section 2.21. 

For sample estimates to be representative of the population we developed the base household-level weights using three main steps:

 base weights?

Thank you for your comment, we now use the word base instead of basis, page 9, paragraph 1 of section 2.21. 

For sample estimates to be representative of the population we developed the base household-level weights using three main steps:

 clarify this sentence - not sure what you're trying to say 

We have clarified this sentence on page 9, paragraph 2 of section 2.21. 

Because there were significant differences between educational attainment by relation to the household head, location, and livelihood in the MLCS, weighting factors for step (3) were calculated based on the share of adults with low education aged 13-69 years old in 2017 (i.e., who would be 18-74 years old in 2022), by relation to the household head (head and spouse, versus other household members), by urban/rural location, and household livelihood within each State or Region.

 can you use a more descriptive, specific word than 'better'? more complete? more comprehensive?

Thank you for your comment, we feel that more comprehensive is the best term. This correct is on page 12, paragraph 1 of section 3.1 

The geographical coverage of the MHWS is more comprehensive in comparison to former national surveys.

 enumeration usually means listed on the sample frame. do you mean that or do you mean interviewed?

We meant interviewed, and we have adjusted the language as such, page 13, paragraph 2 of section 3.1. 

While in both round 1 and round 2 we surveyed 310 townships, in round 3, three additional townships were not interviewed, all of which have very low population sizes.

 this section prob fits better in 'discussion' section 

Thank you for your comment. We have removed this discussion from the results section and moved parts of it to the conclusion, which is our discussion section. 

Time- and cost-considerations are also worth noting here. The survey was designed and implemented in a very short time span (a few months). Moreover, these time advantages offer additional advantages: phone surveys can be implemented at higher frequency, MHWS was implemented four times in 2022. This is particularly advantageous in volatile economic and political settings. Further, face-to-face surveys, such as MLCS, use a two-stage survey design, and cluster typically between 12 and 30 survey households within each enumeration area to reduce transport and other logistical costs. Clustering strategies used in in-person surveys can reduce precision of indicator estimates. These cost savings are necessary for face-to-face surveys but not for phone survey interviews. 

 this is not results. should go in background or discussion. 

Thank you for your comment. We have removed this discussion from the results section as well, and moved parts of it to the conclusion, which is our discussion section. 

 not following these 2 sentences

We have clarified the two sentences as shown below, page 15, paragraph 1 on section 3.2: 

Even though half of the phone survey respondents were female, the phone survey sample consisted of only 6 percent women-adult-only households (WAH) instead of 9 percent as in the MLCS dataset. However, after weighting, the share of WAH households increased to 9 percent, the same as MLCS.

 remind readers length of time btwn r 1 and r4 

We have added the dates of round 1 and round 4 to remind the readers. 

Between round 1 (December 2021- February 2022) and round 4 (October-December 2022)

 what do you mean 'if the survey team could not meet those criteria' ?

We apologize for our lack of clarity we were referring to the quotas. We have now included that language in the sentence on page 19, paragraph 2 of section 3.4. 

If the survey team could not meet the quotas within the township, they called households that met the quotas from the same state/region.

 title of table needs to be improved and clarified - crosstabulation not a title i often see 

We have changed both the title and the contents of the table because of your comment and comments from the other reviewers. 

Table 5. Overview of how often households were interviewed across the four survey rounds, by survey round

# times that sample households were interviewed among the four rounds: Round 1 Round 2 Round 3 Round 4

 N % N % N % N %

One 3,725 30.8% 1,280 10.5% 828 6.8% 3,724 28.8%

Two 2,043 16.9% 2,513 20.7% 2,610 21.5% 1,670 12.9%

Three 1,501 12.4% 3,518 29.0% 3,859 31.8% 2,699 20.9%

Four 4,831 39.9% 4,831 39.8% 4,831 39.8% 4,831 37.4%

Total # households interviewed 12,100 12,142 12,128 12,924 

 just say cell phone surveys you dont have to specify socioeconomic , and the broader history is that cell phone surveys had been building the past 10 years 

We have corrected for the two points. We have also updated the introduction to include this sentiment. 

While the use of cell phone surveys in lower-middle income countries has increased over the last decade, COVID 19 led to a jump in cell phone surveys, but only a few of these aimed for and claimed national representativeness – let alone subnational level representativeness.

Despite their cost-effectiveness and ability to access respondents in hard-to-reach places, prior to COVID-19 the use of national phone surveys was limited in lower-middle income countries, though growing (Dillon 2012, Demombynes et al. 2013, Dabalen et al. 2016, Etang et al. 2020,Maffioli 2020).

 i would argue this is not novel. lau did this, greenleaf (2019, burkina) did this. and there has been more after COVID. what's novel is your estimates seem to be much closer than the aforementioned work in SSA. this is likely bc their is high phone ownership in myanmar. 

We have edited the sentence as follows on page 23, paragraph 2 of section 4:

In this paper, we described an approach to implementing a nationally and sub nationally representative phone survey from a non-representative survey-frame, rather than from a pre-existing survey. The ingredients in this approach were: 

 target-based or quota? be consistent with the terms ou use 

We have changed the language to quota, throughout. The specific example from page 23, paragraph 2 of section 4 is below. 

A quota-based sampling strategy designed to reduce common phone survey biases (such as geographical bias, over-sampling of more educated and urban respondents) and to achieve gender parity as well as an over-sampling of sub-samples of interest (in this case, farm households); 

 you dont need to go into this claim as your paper is not about this and you cover so much else 

We have deleted the following points: 

For example, random digit dialing in step (1) – instead of a phone database with geographical location already known – would require a very large number of phone numbers to be called in order to achieve the targets outlined in step (2); obtaining sufficient numbers of households in small states/regions could be prohibitively costly. Indeed, we roughly estimate that random digit dialing is around twice as expensive as the approach used here.

 you cite a lot of recent work by the world bank but a lot of other groups have been looking into this so i would suggest broadening and updating your search. 

Thank you very much for your comment, we have updated the search to include research by other researchers in the field. Page 30, paragraph 8 of section 4:

There is also some evidence that responses in phone surveys can be systematically different to those of in-person surveys (Lamanna et al. 2019; Greanleaf et al. 2020; Anderson et al. 2023), and that response fatigue may be at least as problematic in shorter phone surveys as it is in longer in-person surveys (Holbrook et al. 2003; Abay et al. 2021) and would also occur in repeated surveys (Schundeln 2018). 

We have added citations from Holbrook et al. 2003, Anderson et al. 2023 and Greanleaf et al. 2020

 this is the first time you use surveillance in the manuscript other than the title. were you doing a survey or conducting surveillance? CDC has a definition of surveillance; i''d see if your work reflects that (or WHO) definition. survey or surveillance? 

We have changed the word to survey from surveillance throughout the manuscript. We have also changed it in our title. 

 and repeated cross-sectional (not sure you can say a panel but maybe you can bc panel is among same group and it is mostly the same group?) 

We are now referring to the survey as a quarterly phone survey. It is not really cross-sectional or a panel, as it is an unbalanced panel. This is now the last sentence of the paper:

Phone surveys can gauge many of the key impacts of these shocks and identify vulnerable households, the effectiveness of their coping mechanisms as well as external interventions, and potentially identify key trends–especially in agriculture–to inform early warning systems.  

Reviewer 2 

Thank you very much for your appreciation of our manuscript. We hope the next draft further clarifies the methods we use for assessing representativeness and weighting. In this revised version, we have endeavored to implement your suggestions and address your concerns, as well as those of the other reviewer and of the editor. Below we explain how we have handled your specific comments. For clarity, we first reproduce your original comments, in bold italics, followed by our response, in roman font, and the new lines in the text in italics. 

 not clear to me and not sure what 8,28 is. From table 

Thank you very much for your kind words and your comment. We have changed Table 5 and updated the table to try to avoid confusion. Please find the new title and table below. 

Table 5. Overview of how often households were interviewed across the four survey rounds, by survey round

# times that sample households were interviewed among the four rounds: Round 1 Round 2 Round 3 Round 4

 N % N % N % N %

One 3,725 30.8% 1,280 10.5% 828 6.8% 3,724 28.8%

Two 2,043 16.9% 2,513 20.7% 2,610 21.5% 1,670 12.9%

Three 1,501 12.4% 3,518 29.0% 3,859 31.8% 2,699 20.9%

Four 4,831 39.9% 4,831 39.8% 4,831 39.8% 4,831 37.4%

Total # households interviewed 12,100 12,142 12,128 12,924 

We have also added two appendix tables on attrition S10 and S11, which show how attrition changes compared to round 1, and an overview of attrition in the sample. 

 

Reviewer 3 

Thank you very much for your careful reading and thoughtful assessment of our initial 

manuscript. In this revised version, we have endeavored to implement your suggestions and 

address your concerns, as well as those of the other reviewer and of the editor. Below we explain 

in detail how we have handled your specific comments. For clarity, we first reproduce your 

original comments, in bold italics, followed by our response, in roman font, and the new lines in the text in italics. 

Major comments:

1. Non-sampling errors in phone surveys. The paper argues that phone surveys should not be considered second-best to in-person surveys suggesting that the phone survey in question outperformed previous in-person surveys. But the paper focuses on sampling design, showing the success of this phone survey in that domain. However, a key advantage (arguably the most important one) of in person surveys is that they allow for longer interviews, with more intensive interviewer supervision. As a result, there seem to be limitations to what data can be collected over the phone and questions about how reliable the data is, i.e. phone surveys may be greater non-sampling error. I see no serious discussion of this aspect. Conclusions about the utility of phone surveys relative to in-person will depend critically on this aspect. Relatedly, it would be good to understand what data items were collected in this phone survey and how reliable we can deem the data. E.g. it appears ‘poverty’ was monitored. Does this mean the survey collected a full consumption module over the phone? How successful was this? Etc. This also speaks to the question if and how phone surveys can be used for welfare monitoring.

Thank you very much for your comments. You are right, and we agree we did not give sufficient attention to this aspect of capturing less information during a phone interview. We only acknowledged the shorter duration of the interview in the previous version of the text. 

To address your points first we toned down the statements that argue that a phone survey is equal to an in-person survey (i.e. we removed the text on it not being a second-best). The intended focus of our paper was on demonstrating a sampling and weighting method to make a phone survey nationally representative, not on proving that a phone survey may be as good as an in-person survey. We have removed all the language that indicates this from the text. 

Further, we now more explicitly discuss the limitations with phone survey length in the abstract, introduction and conclusion section. In addition, we highlight that compared to in-person survey data collection there are likely several potential shortcomings to implementing phone surveys including shortened survey length, enumerator trust, and measurement error. More specifically, we have added the following to page 25, paragraph 8 of section 4. 

Phone surveys also have other limitations. Phone surveys are short-duration interviews with a limited number of questions relative to in-person interviews. The average time for an MHWS questionnaire by phone was 36 minutes, whereas in-person surveys often last three to four times longer. The Myanmar Living Conditions Survey, for example, had an average interview duration of 140 minutes [27]. A setup of rotating modules that are asked only once across different phone survey rounds could try to attenuate such shortcoming, though even across four survey rounds a phone survey is unlikely to capture as much as an in-person survey. Moreover, there is also some evidence that lower enumerator trust and higher measurement error in in phone surveys means that responses in phone surveys can be systematically different from those of in-person surveys [28, 12], and that respondent fatigue may be at least as problematic in shorter phone surveys as it is in longer in-person surveys [26] and would also occur in repeated surveys [29].

Further, we have added a paragraph to the methods section in which we detail more specifically the indicators we collect, and we do not collect. We removed poverty from the text to avoid setting wrong expectations. Indeed, we cannot estimate consumption-based poverty based on our survey questionnaire, which we now highlight in the text. The questionnaire did not contain a full consumption module, it only included indicators associated with poverty, which we agree is not the same. We do feel though that we can use the term welfare more broadly, given that there are a range of other indicators included in the questionnaire that are commonly used in in-person surveys as well, but that are less time-intensive, e.g. related to food insecurity, key household assets, coping, and even happiness and worry. We also include a supplementary appendix in which we highlight what modules we collect in MHWS compared to what modules were collected in MLCS. The tables also include the average time it took respondents to complete the two surveys. We also refer the reader to the survey instruments, so that they can see exactly which questions were asked. Please see the text on page 4, paragraph 1 of section 2. Also please see supplemental tables S1 and S2. 

The objective of the survey was to collect data on a wide range of household and individual welfare indicators–including wealth, unemployment, food security, diet quality, subjective wellbeing, and coping strategies (survey instruments of rounds 1 – 4 can be found in [14]). In the S1 Table we compare the indicators we collected in MHWS to the indicators collected in the 2017 Myanmar Living Conditions Survey (S2 Table), the most recent in-person nationally representative dataset for Myanmar. Because phone surveys are required to be shorter in duration than in-person surveys, MHWS does not collect detailed household member information or detailed food and non-food expenditures. In each round, the same questions were asked to capture changes in the indicators over time.

In response to your concern regarding non sampling error we added a paragraph on the quality control mechanisms in place in the methodology section, page 4, paragraph 3 of section 2.

To minimize measurement error, a quality control team monitored the progress of data collection and was on stand-by to provide feedback to interviewers as needed. Moreover, interviews were recorded, and supervisors were assigned a random selection of one out of five interviews on which to conduct live recording checks. In addition, the data team performed daily checks to screen for outliers and conflicting responses, which were rectified by listening to the audio recordings or by calling back to the respondents for further clarifications. 

2. Attrition: The attrition rates in this survey are severe and so attrition deserves to be discussed in more detail. How does it compare to other phone surveys? How does it compare to in-person surveys? The extent of attrition relative to other types of phone or in-person surveys is a key element in determining which survey mode can be considered reliable or first-best. In light of this, it may not be warranted to conclude that the phone survey in question performed better than other phone surveys – it certainly seems to have succeeded in reaching the targeted respondents but its success also depends on such factors as attrition and non-sampling errors. Relatedly, it would be good to show more clearly the attrition rates e.g. in a table.

This is a very good point. First, we changed the paper in the body of the paper to highlight the number of times each household was interviewed. Further, we added two supplemental tables on attrition. The first shows attrition in a more classical sense treating round 1 as if it was a baseline survey and presenting the percentage of round 1 households that were still interviewed in the subsequent rounds. The second highlights which households participated in which rounds. 

The attrition rates in our survey are severe. However, providing good comparisons is difficult. It was very hard for us to find an example of nationally representative panel in-person or phone survey in an insecure environment like Myanmar. Attrition to some extent can be mitigated by devoting more efforts to tracking respondents. In our survey, we only ask enumerators to call back the same household five times before replacing the household. Furthermore, we ask enumerators to first call back households from the previous round, before calling households from the round before. In this way, we put less effort into preserving our baseline sample, opting for a greater number of within-household comparisons to the previous round instead. This may be something we want to change moving forward.

Nonetheless, we added the following paragraphs in the text to highlight both the high rate of attrition in our sample, but also the lack of in-person counterfactual. Page 19, paragraphs 3, 4, 5, and 6 under section 3.4: 

Thirty-six percent of respondents in round 1 could not be interviewed again in round 2. Yet, some of these were interviewed again in later rounds, thus only 31 percent of round 1 respondents were interviewed only once (Table 5 and S10 Table). Moreover, attrition rate decreased in following rounds. S11 Table shows in detail when households drop out and re-enter the panel. Twenty-five percent of round 2 respondents could not be interviewed in round 3, and only 11 percent of them were interviewed only once across the four survey rounds. Similarly, 24 percent of round 3 re-spondents could not be interviewed in round 4, and only seven percent of round 3 households were interviewed only once (i.e., many of them were interviewed in the preceding rounds 1 or 2). 

Whether our phone survey out- or underperforms in-person surveys in terms of attrition rates is difficult to assess given that we ideally would compare to a survey at a similar frequency (the interval between MHWS survey rounds was only three months), and in a similar setting (heavily impacted by severe shocks). Moreover, attrition can be further mitigated by devoting more efforts to tracking respondents. In our survey, we only ask enumerators to call back the same household five times before replacing the household. 

 In-person interviews are less commonly done in settings that are heavily impacted by severe shocks, and in-person panel surveys are often (though not always) implemented several years apart. In Myanmar’s case, no in-person nationwide panel household surveys were conducted in the past. The Integrated Household Living Conditions Assessment (IHLCA) survey conducted in 2014 and 2019 was only a half-panel, i.e. only half of all households in each survey cluster were re-interviewed whereas the other half of interviewed households newly entered the sample [23]. 

An interesting comparison is a recent study in Malawi and Liberia, where respondents were given cell phones and were administered a phone survey every two months which were initiated prior to, but continued during, the COVID-19 pandemic. [25]. Attrition between rounds was lower in Malawi compared to Liberia, likely because of a better cellular network in the former. In Liberia attrition averaged 49 percent in 2021 and increased over time, as respondents changed phone numbers or opted out of the surveys. When phone surveys were conducted in Ethiopia during conflict, respondents from the severely affected Tigray region were not interviewed at all (25 percent of the sample), but attrition rates among respondents in the other regions were low (1 percent of the remaining sample) [26].

2. Broader policy implications: The paper summarizes the design of a successful phone survey in Myanmar. What are the policy implications and relevance for contexts beyond this specific one? This specific survey relied on a large database from which the survey designers could build their sample – is a phone survey from scratch only possible with a database like this? It is not something survey designers and researchers always have at their proposal. So how does this get replicated? Do NSOs or survey firms start putting together a database? Should this be supported by INGOs/IOs etc.?

Thank you for this question. We have added the following to the conclusion section: 

We demonstrated an effective method for designing and implementing phone surveys that are nationally and sub nationally representative in reaching areas affected by natural or human-made shocks. In doing so, our example from Myanmar serves to encourage researchers to consider phone survey data collection as a different yet effective alternative to in-person surveys. In several contexts, the ability to reach people in areas affected by natural or human-made shocks may outweigh potential disadvantages of phone surveys related to a shorter interview duration, the lack of in-person interactions or others. 

Building a large database of phone numbers to work from (as in our case) is likely a major advantage and is necessary to conduct representative phone surveys. One way to do this is to start with RDD (manually or by using a system) along with continuous snowball sampling recruitment. Another way is hiring local recruiters. Or instead, one could build on the most recent nationally representative in-person survey and call back the participating households with at least one phone number listed [30].

3. Sampling of respondents: The lack of representativeness in other phone surveys (e.g. those discussed in the Brubaker et al. paper that is cited frequently here) is due to the deliberate targeting of a knowledgeable adult in those surveys who can respond on behalf of the entire household. The approach followed in this survey is a convenience sampling approach. Two notes: first, there may be a trade-off here between choosing a reliable respondent and achieving better representativeness. Second, from a sampling standpoint, best practice is a probability selection method rather than selection by convenience. It may be worth discussing these points.

This is a really good point, thanks. We have added more emphasis on this throughout the text. In section 2.1, paragraph 6, we added: 

This approach of respondent selection purposively deviated from the deliberate targeting of a knowledgeable adult, which is commonly done in household surveys. The intention was to ensure a better representation of the entire adult population in our sample, regardless of the persons’ status in the household.

In section 3.3, paragraph 2, we added: 

Our purposive strategy of interviewing adult household members that were not necessarily the most knowledgeable respondent or household heads, ensured to a large extent that our dataset does not suffer the same shortcomings as noted by Gourlay et al. (2021) and Brubaker et al. (2021), where respondents are disproportionally male and household heads.

In section 4, paragraph 5, we added:

When designing the sampling strategy, the survey team purposively chose to focus on a more inclusive survey towards different adults with different roles in the households. This approach contrasts to many data collection efforts which often intentionally aim to interview the household head or the most knowledgeable respondent related to the topic of interest. By doing this we might be trading off accuracy of data about the household at large for better accuracy of data related to individual respondents themselves. 

4. Discussion of weighting methods. These sections should be more detailed and may benefit from a more formal treatment to make it clearer to readers how these methods work.

Thank you for your comments. We have added formulas to section 2.21 for a more formal discussion of how these methods work. The updated discussion is below: 

For sample estimates to be representative of the population we developed the base household-level weights using three main steps:

 Apply an expansion factor: We weight households for their probability of occurring in the sample. This step ensures representativeness at state/region level and the share of households in rural (urban) locations in each of these States and Regions. This can be written as follows: 

〖hh weight v0〗_is= n_(s )/N_s (1) 

In which n_s is the number of households in each strata s (i.e. urban or rural location of each State/Region as in the 2019 ICS) and N_s is the number of observations in each strata.

Adjust for oversampling of farm households: In rural areas of each state and region we proportionally adjust the household weight according to the farm or non-farm status f of households to have the same percentage of farm households as found based on MLCS estimates in each rural strata s as follows:

〖hh weight v1〗_is= 〖hh weight v0〗_is* 〖perc MLCS〗_( fs )/〖perc MHWS 〗_fs if rural=1 (2) 

In (2), 〖perc MHWS 〗_fs was calculated using 〖hh weight v0〗_is . No further correction for livelihoods was made at the urban level given the low number of farmers in that category. 

Weight for education level of the respondent: We proportionally re-weight households to ensure that we achieve a similar percentage perc in each strata s of respondents with education level e (i.e., to adjust for oversampling of more educated respondents), as follows:

〖base hh weight〗_is= 〖hh weight v1〗_( is)* 〖perc MLCS〗_( ehfs )/〖perc MHWS〗_ehfs (3)

Because there were significant differences between educational attainment by relation to the household head, location, and livelihood in the MLCS, weighting factors for step (3) were calculated based on the share of adults with education level e (low education or high education level) aged 13-69 years old in 2017 (i.e., who would be 18-74 years old in 2022), by relation to the household head h (head and spouse, versus other household members), by household farm or non-farm livelihood f, and by strata s (i.e. urban/rural location in each State or Region). Analyses of MLCS show no significant difference between the share of men and women who have low educational attainment, so weighting based on gender of the respondent did not seem warranted. In (3), 〖perc MHWS 〗_ehfs was calculated using 〖hh weight v1〗_is.

 Comparison to in-person surveys: This paper relies on the number of townships reached (or not reached) as a metric of their success. Is this appropriate? A survey may not need to reach all townships to be representative. Specifically, it would be good to understand if the in-person surveys excluded certain townships because those were not selected due to sampling design or because those should have been selected but could not be reached due to insecurity.

Thank you, this is a very important point. It is true that a survey may not need to reach all townships to be representative, but we argue that reaching more township demonstrates a wider geographical spread of data, and therefore likely captures a more diverse selection of respondents. 

We have updated the language in page 12, paragraph 1of section 3.1 as follows: 

The geographical coverage of the MHWS is more comprehensive in comparison to former national surveys. We use townships as an indicator of the geographical spread of the data.

We have also added text to clarify both in the below but also in the conclusion that MLCS and other in-person surveys did not enumerate important townships due to conflict. Page 14, paragraph 6 under section 3.1:

In MLCS, because of security and access problems, 33 enumeration areas had to be replaced and seven were not visited at all. This includes three townships in Rakhine state that could not be interviewed due to security concerns over the 12-month enumeration period. 

It should be noted that MHWS and MLCS did not intend to interview 19 townships that were 

 Introduction: Phone surveys were limited in low-income contexts but very common in Europe and North America for instance.

Thank you for your comment, this is an important point, we have added this in line 1, as well as the title of the paper. We now try to highlight that the use of phone surveys was common in Europe and North America.

For decades, in-person data collection has been the standard modality for nationally and sub-nationally representative socio-economic survey data in low- and middle-income countries.

 Incentives: Clarify if any incentives were used

Thank you for your comment, we have added the following to page 4 of paragraph 2 of section 2. 

MSR initially provided respondents the equivalent of $2.20 per round as an incentive (Round 1 and round 2) but increased that to the equivalent of $3.50 per round for subsequent rounds. The incentive is provided as a phone credit to the phone number that is contacted, using online systems for mobile phone credit purchases. 

 -S6 Table shows that differences between adult respondents in the MHWS and the adult population in the MLCS further reduce after weighting the MHWS data rather than aggravate as in Brubaker et al. (2021).” It is my reading of the Brubaker et al paper that they find that sampling weights reduce rather than aggravate differences, but do not reduce enough to overcome significant differences. 

Thank you for this comment. We re-read the Brubaker et al. (2021) study, and it seems they found mixed results. On page 14, they write: 

"The effectiveness of the bias reduction is mixed and depends on the outcome of interest. Compared to the estimates obtained under the HFPS household sampling weights, the estimates based on the HFPS individual weights move closer to those for the general adult population for most individual-level outcomes of interest. However, confidence intervals widen as well. Several points stand out.

First, there are instances where the HFPS household weight (w1) increases the difference between the unweighted respondent data and our benchmark weighted F2F survey sample. Notably, the incidence of headship moves further from the mean in all four countries, though the difference is easier to detect in Nigeria and Uganda. […]”

We therefore adjusted the text as follows on page 17, paragraph 3 under section 3.3: 

In contrast, Brubaker et al. (2021) found mixed results due to weighting, with some individual-level outcomes getting closer to the benchmark population but others worsening.

---

## [Decision Letter · Decision Letter 1]

13 Nov 2023

PONE-D-23-25141R1Can phone surveys be representative in low- and middle-income countries? An application to MyanmarPLOS ONE

Dear Dr. Van Asselt,

Thank you for submitting your manuscript to PLOS ONE. After careful consideration, we feel that it has merit but does not fully meet PLOS ONE’s publication criteria as it currently stands. Therefore, we invite you to submit a revised version of the manuscript that addresses the points raised during the review process.

In particular, I believe that the manuscript could be improved through minor revisions in response to the comments of reviewers 1 and 3, and please thoroughly check English grammar and typographical errors.

We look forward to receiving your revised manuscript.

Kind regards,

Seo Ah Hong, PhD

Academic Editor

PLOS ONE

Journal Requirements:

Reviewers' comments:

Reviewer's Responses to Questions

**Comments to the Author**

1. If the authors have adequately addressed your comments raised in a previous round of review and you feel that this manuscript is now acceptable for publication, you may indicate that here to bypass the “Comments to the Author” section, enter your conflict of interest statement in the “Confidential to Editor” section, and submit your "Accept" recommendation.

Reviewer #1: All comments have been addressed

Reviewer #2: All comments have been addressed

Reviewer #3: All comments have been addressed

2. Is the manuscript technically sound, and do the data support the conclusions?

Reviewer #1: Yes

Reviewer #2: Yes

Reviewer #3: Yes

3. Has the statistical analysis been performed appropriately and rigorously? 

Reviewer #1: N/A

Reviewer #2: Yes

Reviewer #3: Yes

4. Have the authors made all data underlying the findings in their manuscript fully available?

Reviewer #1: Yes

Reviewer #2: Yes

Reviewer #3: Yes

5. Is the manuscript presented in an intelligible fashion and written in standard English?

Reviewer #1: Yes

Reviewer #2: Yes

Reviewer #3: Yes

6. Review Comments to the Author

Reviewer #1: Thank you for your thorough responses to my 47 (!!) comments.

Along with the feedback and updates suggested by Reviewer 3, the manuscript is much improved.

The remaining piece is to improve the discussion section.

Two comments on this:

1 - in the results section, you added new information about attrition. it is a great add but you have discussion section information there - starting with "Whether our phone survey out- or ... but attirition rates among respondents in the other regions were low (1 percent of hte remaining sample) [28]." belongs in discussion section

2 - discussion section should be shaped as follows:

·  Summarize your findings (Give the key results that have lessonslearning for the others to replicate.)· 

 Studies that are consistent or in contrastwith your study· 

 What your study adds·  

Strengths and limitations

·  Conclusion

Reviewer #2: The authors have addressed my questions earlier and I am happy with the response. Thank you for the great work.

Reviewer #3: I commend the authors for thoroughly addressing my comments. I am recommending to accept this paper.

I have two minor follow-up comments for the authors, which however do not require a formal revision. If the authors see fit they can incorporate these comments.

First, on attrition. I believe the Gourlay et al. paper has some information on attrition for the series of surveys they cover, which may be worth referencing.

Second, on weighting. I share the general conclusion of the authors that weighting can only do so much to improve representativeness and it is also fair to say that weighting has somewhat mixed results especially if initial differences are large. However, the Brubaker et al paper does not entirely support this conclusion. I went back to the paper, and my read is that the authors are referring to the fact that household weights (w1) can make things worse, while individual weights (w2) at least do not make things worse. This is my read of this figure: https://journals.plos.org/plosone/article/figure?id=10.1371/journal.pone.0258877.g001

I wish the authors all the best in completing this process.

7. PLOS authors have the option to publish the peer review history of their article (what does this mean?). If published, this will include your full peer review and any attached files.

Reviewer #1: No

Reviewer #2: No

Reviewer #3: No

---

## [Author Response · Author response to Decision Letter 1]

1 Dec 2023

December 1, 2023 

Ref: Resubmission of Manuscript PONE-D-23-25141

Dear Editor:

We appreciate the opportunity to revise our research manuscript " Can phone surveys be representative in low- and middle-income countries? An application to Myanmar.” We revised the document according to your recommendations and the reviewers’ thoughtful comments. Below we explain in detail how we have handled the specific comments. For clarity, we first reproduce the original comments, in bold italics, followed by our response, in roman font, and the new lines in the text, in italics. 

Reviewer 1

Thank you for your thorough responses to my 47 (!!) comments.

Along with the feedback and updates suggested by Reviewer 3, the manuscript is much improved. 

The remaining piece is to improve the discussion section.

Two comments on this:

1- in the results section, you added new information about attrition. it is a great add but you have discussion section information there - starting with "Whether our phone survey out- or ... but attirition rates among respondents in the other regions were low (1 percent of hte remaining sample) [28]." belongs in discussion section

Thank you very much for your commitment to improving the quality of our paper. We have made the following changes in response to your comments. We have added a discussion section and now placed the discussion on attrition in the discussion section as part of the discussion of studies that are consistent or contrast with our study. 

2- discussion section should be shaped as follows:

· Summarize your findings (Give the key results that have lessonslearning for the others to replicate.)· 

 Studies that are consistent or in contrastwith your study· 

 What your study adds· 

Strengths and limitations

· Conclusion

We have added a discussion section and followed the format that you have recommend. First, we highlight our findings and the key results for other authors to replicate (page 23 paragraph 1). Next, we highlight other studies that are consistent or in contrast with our study and emphasize what our study adds to the field on pages 24 and 25. This is followed by a discussion of strengths and weaknesses on page 27. We hope this new format will make the discussion clearer and more thorough. 

Reviewer 3 

I commend the authors for thoroughly addressing my comments. I am recommending to accept this paper.

Thank you very much for your thorough review of our paper, and continued commitment to improving the quality of our interpretation of the literature. 

I have two minor follow-up comments for the authors, which however do not require a formal revision. If the authors see fit they can incorporate these comments.

1-First, on attrition. I believe the Gourlay et al. paper has some information on attrition for the series of surveys they cover, which may be worth referencing.

Thank you for pointing this out, we have now included the following lines in our discussion section.

When comparing the response rate to the COVID-19 high frequency surveys implemented among participants in recent in-person Living Standard Measurement Surveys (LSMS) in five different countries in Sub-Saharan Africa, as many as 94 percent of households with phone numbers were reached in Uganda but only 62 percent Ethiopia [10].

2-Second, on weighting. I share the general conclusion of the authors that weighting can only do so much to improve representativeness and it is also fair to say that weighting has somewhat mixed results especially if initial differences are large. However, the Brubaker et al paper does not entirely support this conclusion. I went back to the paper, and my read is that the authors are referring to the fact that household weights (w1) can make things worse, while individual weights (w2) at least do not make things worse. This is my read of this figure: 

We are sorry for overlooking this, we now understand that Brubaker et al. improve their individual-level estimates if they use the individual-level weights, which are propensity score-adjusted individual weights. We improved our text on page 19 to highlight this point. 

In contrast, such approach was not satisfactory for individual representativeness in several COVID-19 phone surveys in Sub-Saharan Africa [7], and further recalibration of phone survey sampling weights with propensity score adjustments (based on the individual’s likelihood to be interviewed) was necessary to ensure that their individual-level characteristics would be closer to those of the benchmark population.

---

## [Decision Letter · Decision Letter 2]

10 Dec 2023

Can phone surveys be representative in low- and middle-income countries? An application to Myanmar

PONE-D-23-25141R2

Dear Dr. Van Asselt,

We’re pleased to inform you that your manuscript has been judged scientifically suitable for publication and will be formally accepted for publication once it meets all outstanding technical requirements.

Kind regards,

Md Nazirul Islam Sarker, PhD

Academic Editor

PLOS ONE

Additional Editor Comments (optional):

The author is advised to keep in touch with the production team for the remaining process.

Reviewers' comments:

Reviewer's Responses to Questions

**Comments to the Author**

1. If the authors have adequately addressed your comments raised in a previous round of review and you feel that this manuscript is now acceptable for publication, you may indicate that here to bypass the “Comments to the Author” section, enter your conflict of interest statement in the “Confidential to Editor” section, and submit your "Accept" recommendation.

Reviewer #1: All comments have been addressed

Reviewer #3: All comments have been addressed

2. Is the manuscript technically sound, and do the data support the conclusions?

Reviewer #1: Yes

Reviewer #3: Yes

3. Has the statistical analysis been performed appropriately and rigorously? 

Reviewer #1: Yes

Reviewer #3: Yes

4. Have the authors made all data underlying the findings in their manuscript fully available?

Reviewer #1: Yes

Reviewer #3: Yes

5. Is the manuscript presented in an intelligible fashion and written in standard English?

Reviewer #1: Yes

Reviewer #3: Yes

6. Review Comments to the Author

Reviewer #1: Thank you for re-arranging the results section so that there is no discussion. The discussion section is still a bit long and repetitive - usually discussion section is a bit shorter however I think this is fine. Good luck and thank you.

Reviewer #3: I thank the authors for incorporating my additional comments. I do not have any further revisions to request.

7. PLOS authors have the option to publish the peer review history of their article (what does this mean?). If published, this will include your full peer review and any attached files.

Reviewer #1: No

Reviewer #3: No

---

## [Editor Report · Acceptance letter]

14 Dec 2023

PONE-D-23-25141R2 

PLOS ONE

Dear Dr. Van Asselt, 

I'm pleased to inform you that your manuscript has been deemed suitable for publication in PLOS ONE. Congratulations! Your manuscript is now being handed over to our production team.

Kind regards, 

on behalf of

Dr. Md Nazirul Islam Sarker 

Academic Editor

PLOS ONE